# Multilayer Perceptron and Their Comparison with Two Nature-Inspired Hybrid Techniques of Biogeography-Based Optimization (BBO) and Backtracking Search Algorithm (BSA) for Assessment of Landslide Susceptibility

Hossein Moayedi [1,2,*], Peren Jerfi Canatalay [3], Atefeh Ahmadi Dehrashid [4,5,*], Mehmet Akif Cifci [6], Marjan Salari [7] and Binh Nguyen Le [1,2]

1 Institute of Research and Development, Duy Tan University, Da Nang 550000, Vietnam
2 School of Engineering and Technology, Duy Tan University, Da Nang 550000, Vietnam
3 Department of Computer Engineering, Faculty of Engineering, Haliç University, Istanbul 34394, Turkey
4 Department of Climatology, Faculty of Natural Resources, University of Kurdistan, Sanandaj 6617715175, Iran
5 Member of Department of Zrebar Lake Environmental Research, Kurdistan Studies Institute, University of Kurdistan, Sanandaj 6617715175, Iran
6 Department of Computer Engineering, Bandirma Onyedi Eylul University, Balikesir 10200, Turkey
7 Department of Civil Engineering, Sirjan University of Technology, Sirjan 7813733385, Iran
* Correspondence: hosseinmoayedi@duytan.edu.vn (H.M.); atefeh.ahmadi@uok.ac.ir (A.A.D.)

**Abstract:** Regarding evaluating disaster risks in Iran's West Kurdistan area, the multi-layer perceptron (MLP) neural network was upgraded with two novel techniques: backtracking search algorithm (BSA) and biogeography-based optimization (BBO). Utilizing 16 landslide conditioning elements such as elevation (aspect), plan (curve), profile (curvature), geology, NDVI (land use), slope (degree), stream power index (SPI), topographic wetness index (TWI), rainfall, and sediment transport index (STI), and 504 landslides as target variables, a large geographic database is constructed. Applying the techniques mentioned above to the synthesis of the MLP results in the suggested BBO-MLP and BSA-MLP ensembles. As accuracy standards, we benefit from mean absolute error, mean square error, and area under the receiving operating characteristic curve to assess the utilized models, we have also designed a scoring system. The MLP's accuracy increases thanks to the application of the BBO and BSA algorithms. Comparing the BBO with the BSA, we find that the former achieves higher average MLP optimization ranks (20, 15, and 14). A further finding showed that the BBO is superior to the BSA at maximizing the MLP.

**Keywords:** landslides susceptibility assessment; multilayer perceptron; BBO algorithm; BSA algorithm

## 1. Introduction

Among the most significant environmental risks in the world these days are landslides, having both human and fiscal consequences [1]. Of the most challenging tasks in geological engineering is predicting the movement of landslides in hilly and reservoir environments [2]. Natural calamities are only one of many potential causes of a landslide catastrophe. In addition to the geological circumstances in which they occur, several additional factors may cause landslides to develop [3]. Predicting the deformation and development of landslides [1] is one of the most difficult and crucial challenges in geomorphology [4]. Deformation monitoring and prediction of landslide disasters may lessen the danger of landslides to human populations, property, and infrastructure by understanding the instability process [2] and altering landslide characteristics [5]. Natural hazard studies have a serious problem in predicting landslide susceptibility. These problems manifest in landslides [3] since they are complicated, dynamic [4], and unpredictable systems [5]. Various reasons, including geological, hydrological [6], morphological, and

human-induced [7], might be accountable for the movement they create [6,7]. Various ways of predicting landslides have been developed and deployed in recent years, which may be split into two main groups: quantitative and qualitative methods [8]. In this complicated geological environment [8], conventional methods for predicting total landslides are inadequately exact [9]. Various models, along with the landslide movement prediction technique of analysis [10,11], frequency ratio [12], weights of evidence [13–15], the logistic regression model [16], MCDM models [17], and neural network [18] were used. A mathematical, statistical [9], nonlinear theoretical [10], and complete model followed the initial empirical model for predicting landslide susceptibility. This progression has taken place during the previous 50 years [19]. With the ongoing updating of artificial intelligence systems, certain nonlinear landslide susceptibility analysis models have been built [11–17].

Due to the model's high complexity, researchers tend to provide an optimized solution. Optimization is finding the optimum values of a problem's variables to minimize or maximize an objective function. In other words, optimization is finding the best solution to a trial by adjusting the importance of the variables that impact the result [18]. Optimization aims to minimize or maximize an objective function, a mathematical expression that reflects the item being improved. For example, suppose a company wants to make the most money possible. In that case, the objective function could be a mathematical model of the company's profit based on many factors, such as production levels, prices, and advertising costs. By adjusting these variables and finding the values that maximize the objective function, the company may find the most feasible solution to the problem of maximizing profits.

Due to the improvement of numerous solutions during optimization, multi-solution-based algorithms have a more significant local optimum avoidance by nature. In this case, more solutions may allow a solution trapped in a local optimum to escape from it. Multiple-solution-based algorithms examine a more significant section of the search region than single-solution-based algorithms; hence, the likelihood of acquiring the global optimum is greater [18,19]. In addition, information about the search space may be shared across several solutions, which expedites progress toward the ideal. Despite their many benefits, multi-solution-based algorithms need additional function assessments. The most popular single-solution-based algorithms are hill climbing and simulated annealing. Both algorithms are based on a similar premise, but stochastic cooling allows SA to avoid local optimums more effectively [18].

Iterated Local Search (ILS) [20] and Tabu Search (TS) [21] are two modern algorithms based on a single solution. Popular multi-solutions-based algorithms [21–23] include genetic algorithms (GA), particle swarm optimization (PSO), ant colony optimization (ACO), and differential evolution (DE). Darwin's evolution of natural selection impacted the design of the GA algorithm. This algorithm sees solutions as solutions, and their parameters reflect their DNA. This algorithm is primarily motivated by natural selection, with the best individuals preferring to contribute more to improving mediocre solutions.

The GA algorithm represents solutions to a problem as "individuals" in a "population", with the parameters of each individual (its "genes") reflecting different components of the solution. The GA algorithm employs concepts that imitate the process of natural evolution, such as selection, crossover (recombination), and mutation, to a population of individuals. A genetic algorithm seeks to find the ideal solution to a problem by "evolving" a population of solutions over time. The concept of "survival of the fittest" is implemented in the GA algorithm via the selection process, in which the best individuals have a greater chance of being picked to participate in the evolution of the population. This guarantees that the population improves over time since the best solutions will likely be passed on to future generations. The PSO algorithm resembles the feeding behavior of flocks of birds and schools of fish. This algorithm improves solutions relative to the best solutions previously reached by each particle and the best solution improved by the "swarm". The ACO algorithm imitates the collective behavior of ants in finding the shortest route from the nest to the food source. DE employs simple formulae that incorporate

the parameters of previous solutions to expand the candidate pool for a specific problem. Two features distinguish the two types of nature-inspired algorithms [24]: enhancing solutions until they fulfill end criteria and splitting the optimization process into two parts, exploration and exploitation. Exploration is an algorithm's propensity to display highly unexpected behavior, resulting in significantly different solutions. Significant differences in the solutions motivate a deeper exploration of the search space and, therefore, the identification of its most promising sections. As an algorithm leans toward exploitation, solutions often encounter changes on a smaller scale and prefer to explore them locally. A combination of exploration and exploitation may lead to discovering the optimal global solution for a particular optimization problem [18].

The conventional numerical methods mainly include catastrophe theory [20], PSO neural network, cooperative work theory [21], back propagation neural network model [22], support vector regression model [23], and chaos model (Huang et al., 2018), decision tree model [24,25], long short –term memory [26], neural network extreme machine learning model [27], Elman neural network model [25] and so on. A combination method has been applied to attain efficient outcomes in the current work [26]. In this regard, researchers are now using meta-heuristic strategies to improve efficiency due to the limitations of current models, including local minimum and dimension dangers [27]. In this respect, all of these meta-heuristic approaches have a great capacity to resolve optimization issues, and for such reason, they have indeed been implemented in several scientific disciplines. The algorithms have several characteristics, and the majority are population-based techniques. Throughout the calculations, we could perhaps find the best design for each of them. It might be beneficial to create a novel technique that enhances the process or outcomes of optimization.

These techniques are used to find high-quality solutions that are based on the best possible computing structure [28]. Several advanced strategies (including parallel computation, multi-agent systems, and decomposition of the search space) [29] are often used in hybrid metaheuristic algorithms. The problems are solved collaboratively by a proactive search agents group acting individually and with parallel computation. They solved many large-scale distributed and dynamic systems with successful results [30]. Previous studies have shown that not estimating the participation of each parameter in the classification by the optimized ANN model is one of the primary challenges of neural network model optimization algorithms. It also has some limitations and drawbacks, including high computational power requirements and a significant computation time for determining the final result. In cases where immediate results are required, both weaknesses can be problematic [31]. Therefore, in order to have an idea of how effectively different algorithms work, this research was conducted. This research applies optimization methods with the neural network model [32] for comparison methods. The complete optimization methods discussed in this sector work quite well [33]. Researchers have found that the hybrid landslide prediction and zoning method yields the most reliable and productive results. Therefore, the combined methods employed comprise multilayer perceptron and two nature-inspired hybrid techniques of biogeography-based optimization (BBO) and backtracking search algorithm (BSA). This research is one of the first practical studies that attempt to analyze and enhance the outcomes of the neural network model in conjunction with optimization algorithms using a novel strategy. Its findings may be beneficial in natural disaster management.

## 2. Review of Case Study

In western Iran, on the border with Iraq, lies the 280203-square-kilometer Kurdistan province, which accounts for around 1.7% of Iran's total land area (between $34°45'$ and $36°28'$ north latitude and $45°34'$ and $48°14'$ east longitude). Its position relative to the Greenwich meridian is measured in minutes. This study's area encompasses the western, central, and southern half of the province (Figure 1). The majority of this province consists of mountains, and the number of peaks in this area is very high. The average mountain

slope is 30–100%, and the mountains range in height from 1700–2300 m. In general, most of the geological formations in this area have evolved in the second geological epoch. The area under study is located in the Sanandaj-Sirjan tectonic zone. The average annual rainfall in this province varies from around 350 mm to more than 1000 mm, and its climate is Mediterranean. When landslides are most common, March and April have the most rainfall annually. Topography and air currents considerably impact the province's yearly average temperature, which varies from roughly 2 degrees Celsius at heights to about 15 degrees Celsius in low-lying regions and plains.

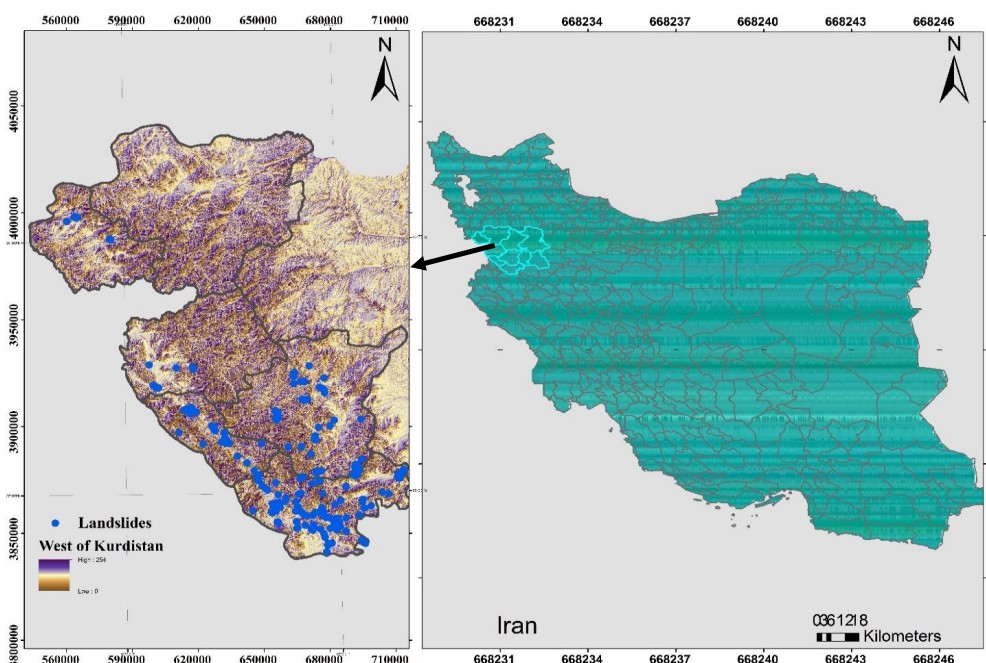

**Figure 1.** The study area of the research in the west of Iran (UTM coordination system).

## 3. Methodology

Maps, modelling methods, validation of the modelling methods, and analysis of the optimization algorithms all need to be completed in order to complete the three tasks outlined above. Detailed explanations of these steps are provided below.

### 3.1. Artificial Neural Network

ANN is an algorithm patterned after biological nerve cells (neurons) structure and can learn independently [34]. Numerous applications, including robotics, pattern recognition, medicine, power systems, signal processing, prediction, and system modeling, make use of ANN. Figure 1 depicts a basic example of ANN [35]. ANN comprises neurons interconnected by synaptic connections. In addition, it acquires new knowledge by drawing conclusions or generating generalizations from the diverse sample data provided to it. The ANN technique is utilized to solve nonlinear issues. Data is separated into a training set and a test set. Adjusting the weights of the neural network in order to lower or minimize the error rate is the objective of the training procedure. This procedure will continue until the desired result is attained. The degree of performance of the training process is determined by evaluating data not utilized during training in the neural network [26–30]. Feedforward backpropagation is an effective design for training neural networks. The feedforward neural network moves from the input layer of the model to the output layer in a single direction. By simulating the human brain's learning process, artificial neural networks (ANNs) are computer software that performs core functions such as producing new data from the data obtained by the brain via learning, remembering, and generalizing. Consequently, of numerical simulations of the learning process, which was inspired by the human brain,

artificial neural networks have emerged. Parallel distributed networks, connected networks, and neuromorphic networks are different names for artificial neural networks.

Two modules comprise the mathematical model of an artificial neuron: (1) linear activation and (2) a nonlinearity that confines signal levels within a specific range. Presented is the structure of a convolutional neural network architecture [36]. For instance, the aggregation function represents the cell body, and the inputs represent the dendrites [37]. The synapse is referred to as an activation function, a nonlinear function. Similar to an axon, the link between the network and the nonlinear unit exists.

The ANN technique is effective with photos, text, and data tables. When dealing with nonlinear functions and learning weights, having the benefit of being able to transform any input into an output successfully ANN is the best technique. Thanks to the activation nonlinear structure the ANN can master any convoluted connection between input and output data, known as a universal approximation. ANNs are being used in the scientific sector [28,29].

The ANN approach uses photos, text, and data tables [30,31]. Figure 2 depicts the architecture of the artificial neural network [38]. Every bias weight is composed of the sum of all neuron signals. All input weights are added to each neuron's output. Backpropagation is beneficial to alter the weights of a neural network in order to obtain gradients. During backward propagation, the gradient may completely vanish or expand.

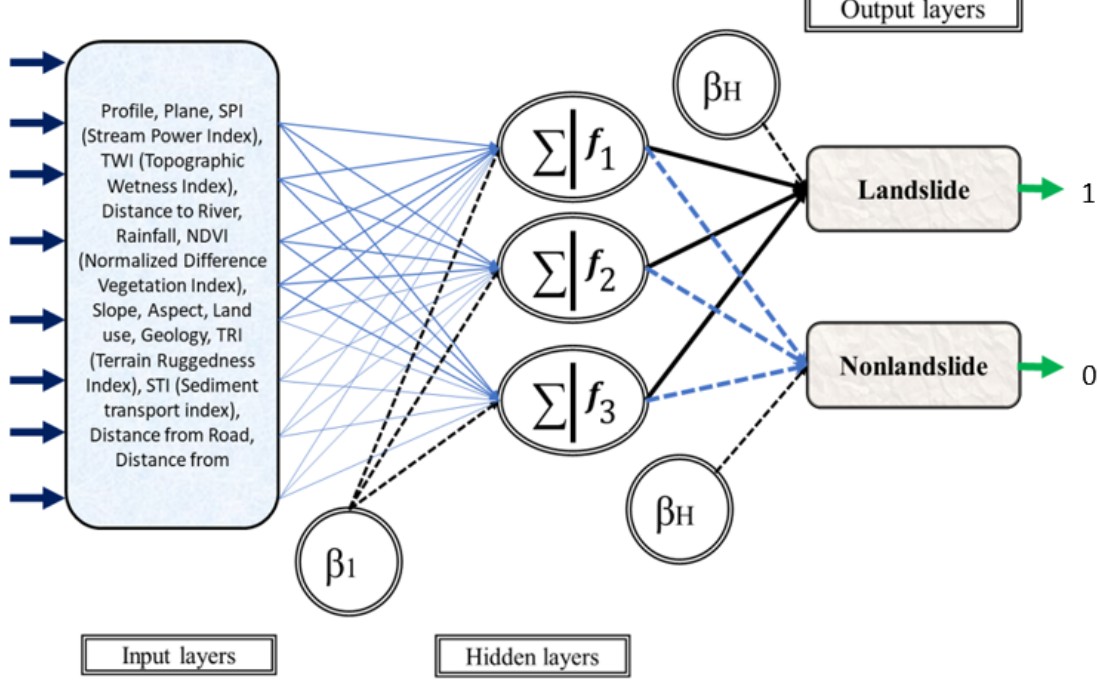

**Figure 2.** Basic structure of neural network for landslide susceptibility analysis with optimization algorithms.

### 3.2. Hybrid Model Development

Figure 3 of the study of the flowchart displays the input parameters for predicting the output (the success rate of landslide susceptibility mapping prediction) [38]. The hybrid models used in this experiment combined the BBO and the BSA. ANNs are used in metaheuristic algorithms [39]. In these combinations, optimization methods are used instead of the Levenberg-Marquardt (LM) method [40], which is usually used for training. Multiple steps comprise this procedure, including:

(a)  Determining the ideal structure of the ANN model: we are aware that the structure of the ANN has a significant impact on the accuracy of its predictions [41]. As the model's backbone, it should be optimized in hybrid models [39]. The network with several processors in the intermediate layer and a Tansig activation function is the

optimal solution, as determined by a trial-and-error procedure applied to various tested configurations.

(b)     Specify the problem function and use the BBO-MLP and BSA-MLP models.
(c)     Specify precise parameters such as population size, number of iterations, and goal function.
(d)     Minimizing inaccuracy by modifying the ANN's weights and biases
(e)     Storing the optimum solution when a termination condition is met.

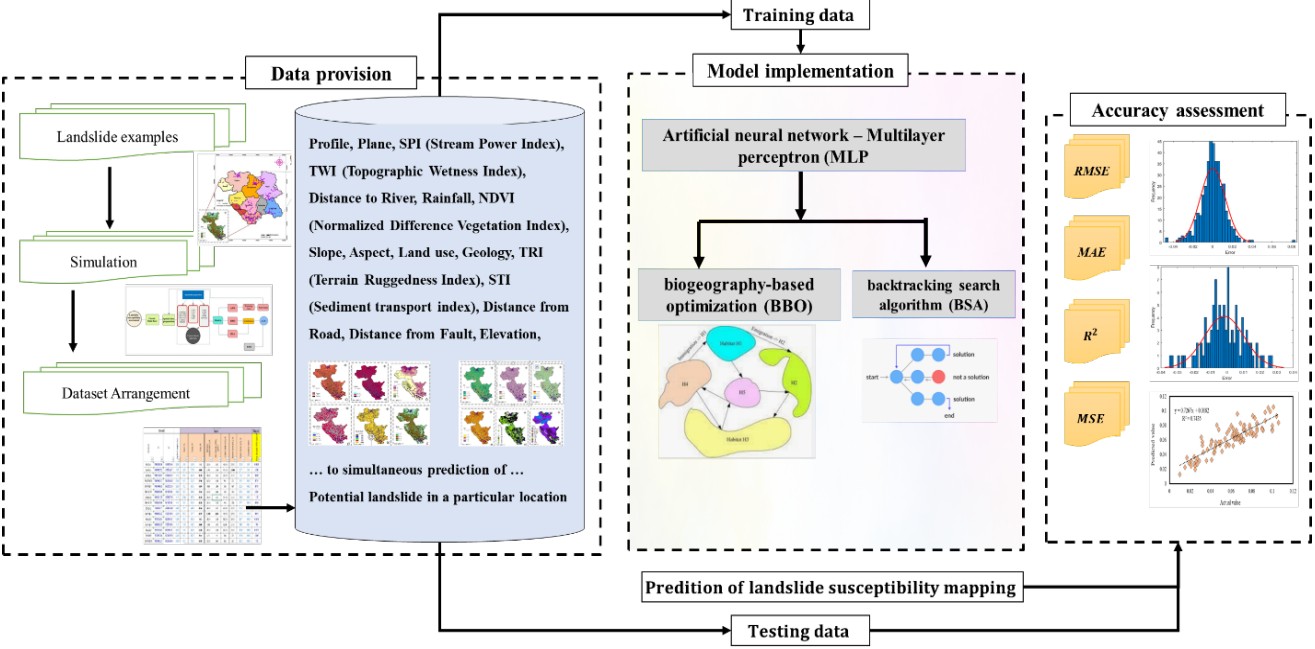

**Figure 3.** Research flowchart including input parameters to predict the output.

In order to find the minimum error (e.g., finding the best predictive network) during the new model evaluation, we used the term mean square error (here abbreviated as MSE) as the objective function. It indicates the quality of each iteration's resulting solution. In Equation (1), the MSE is given as a formula.

$$MSE = \frac{1}{U} \sum_{i=1}^{U} \left| S_{i_{observed}} - S_{i_{predicted}} \right| \tag{1}$$

$S_{i_{observed}}$, and $S_{i_{predicted}}$ display the expected and actual production figures. $U$ also represents the quantity of samples.

In this study, the iteration number is set to 1000. The calculated RMSEs or MSEs are shown as a convergence curve that illustrates the optimization behavior of the architecture given the number of iterations. These curves are shown in Figure 3 for the nine tested population amounts: from 500 to 50 with 50 intervals such as ([500, 450, 400, 350, 300, 250, 200, 150, 100, and 50]). This approach ensures the problem is fixed using techniques of uniform complexity.

### 3.2.1. Biogeography-Based Optimization (BBO)

Using biogeography as a foundation, the BBO algorithm attempts to map the geographic distribution of living organisms. In the 1960s, [42] developed mathematical models of biogeography. They concentrated primarily on the distribution of species inside and between the surrounding ecosystems, as well as the migration of species between habitats. There has been a great increase in the study of biogeography since then. It was not until 2008 that Simon developed a general-purpose optimization algorithm [43]. Listed here are

its BBO steps. In the first step of the BBO, a random population is produced and given the label 'habitat,' similar to that in prior evolutionary techniques [44,45]. The habitat suitability index (HSI) and the suitability index variable (SIV) are used to assess the suitability of these individuals (i.e., possible solutions). Two of the BBO's first functions, migration and mutation, are detailed as follows:

As a first step, feasible solutions are refined to achieve higher levels of quality. In order to establish whether or not SIV adjustments are required, an immigration rate ($\lambda_g$) is calculated [46]. Emigration rates ($\mu_g$) are established in situations where modifications are needed. On a probabilistic basis, it is utilized to figure out which solution to migrate. It excludes highly-fitting solutions from consideration in order to prevent random corruption [44,47].

There are many effective risks in a region. For this reason, making a series of changes in the amount of HIS is not out of mind and its balance may be abnormal. In these cases, a factor is evaluated for each of the population relations in the mutation or non-mutation state, and this factor determines the solution to the existing problems. Therefore, the higher the probability, the higher the accuracyIf S represents the number of species, then the mutation rate is shown as the following equation:

Equation (2):

$$P_g^f = \begin{cases} -\left(\lambda_g + \mu_g\right)P_g + \mu_{g+1}{}^{P_{g+1}} \\ -\left(\lambda_g + \mu_g\right)P_g + \lambda_{g-1}{}^{P_{g-1}} + \mu_{g+1}{}^{P_{g+1}} \qquad 1 \le S \le S_{max} - 1, \\ -\left(\lambda_g + \mu_g\right)P_g + \lambda_{g-1}{}^{P_{g-1}} \end{cases} \tag{2}$$

The subsequent approach depicts the conventional BBO algorithm introduced by Simon (Simon, 2008; Lim et al., 2016).

A Classical BBO Algorithm

(1) BBO parameters require configuration, which consists of emanating a representative method for habitats, that is concerned with hanging and initializing the highest migration rate, transformation rate, and elitism parameter.
(2) Create a random set of habitats based on the possible solution sets and initialize them.
(3) Each habitat's migration and emigration rates can be determined by utilizing its HSI.
(4) Migrate in a random fashion to change the environment of each special habitat. The HSIs were then computed again.
(5) Each habitat should be assigned a mutation rate based on the number of species present.
(6) Random mutations should be applied to every non-light habitat. It was then recalculated for each individual HSI.
(7) To begin the next iteration, go to step one (3). Repeat for as many generations as necessary till you have come up with the right answer.

3.2.2. Backtracking Search Algorithm (BSA)

BSA is a new evolutionary algorithm based on populations [48]. In an iterative process, the objective function is reduced to the minimum possible value. BSA has five evolutionary mechanisms: initialization, selection-I, mutation, crossover, and selection-II. As depicted in Figure 4, the BSA's general flowchart is shown. It is explained in more detail in the following sections.

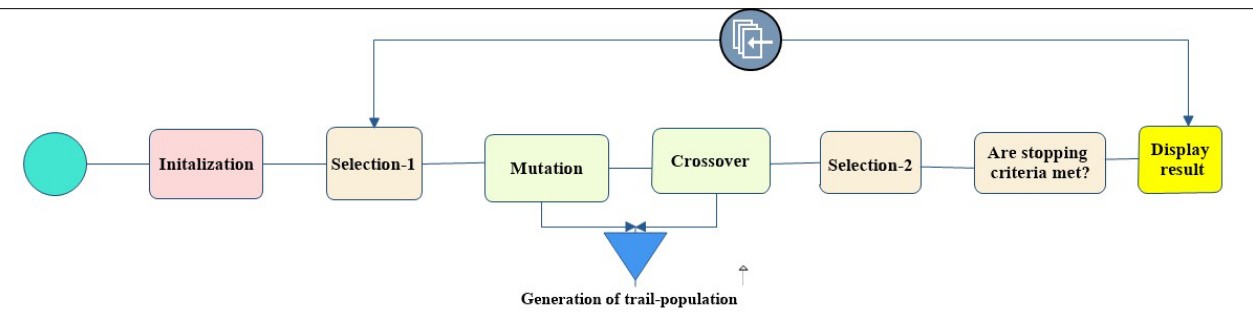

**Figure 4.** General flow chart of BSA.

**A. Initialization:** BSA's initial population (*P*) consists of *D* variables and *N* individuals, which are generated at random. Equivalently, Equation (3) states it this way:
Equation (3):

$$P_{ij} \sim U(low_j, \, up_j). \tag{3}$$

For *I* = 1, 2, ... *N* and *n* = 1, 2, ... , *D*, where N is the population proportion, *D* is the tribulation of dimension, and U is the uniform distribution.

**B. Selection-I:** Pre- and post-selection are handled by the selection-I and selection-II operators, respectively, in BSA. Using the pre-selection operator, the historical population ($P^{old}$) is acquired, and this data is used to decide the search's direction. To find out what ($P^{old}$) is valued, you must perform the following three steps:
Equation (4) is used to randomly construct the first historical population.
Equation (4):

$$P_{ij}^{old} \sim U(low_j, \, up_j) \tag{4}$$

$P^{old}$ is recalculated at the beginning of each iteration by applying Equation (5)
Equation (5):

$$if \; a < b \; then \; P^{old} := {}^{P}/_{a}, \; b \sim U(0,1). \tag{5}$$

where := is the update procedure, and *a* and *b* are random numerals.

Using Equation (6), we can rearrange the individuals of the historical population as follows:
Equation (6):

$$P^{old} := Permuting\left(P^{old}\right) \tag{6}$$

The permutation function is a random shuffling function.

**C. Mutation.** The mutation is applied to the first trial population (mutant) by BSA as described in Equation (7).
Equation (7):

$$Mutant = P + F \cdot \left(P^{old} \cdot P\right) \tag{7}$$

where *F* controls the amplitude of the search direction matrix ($P^{ald-p}$).

**D. Crossover.** As a result of BSA's crossover, the final trial population (T) is formed. Two steps are involved:

1. For the trail population T, *a* binary-valued matrix (matrix) is constructed with a size of *N* × *D*, where *N* is the number of individuals.

2. *n* ∈ {1,2, ... , *N*} and *m* ∈ {1, 2, ... , *D*} are the initial values of the binary integer matrix. The value of T is updated by using Equation (8) as follows:
Equation (8):

$$T_{n,m} := P_{n,m} \tag{8}$$

**E. Selection-II.** BSA greedy selection is the name given to selection-II. The trail population T is replaced if its fitness values outperform those of the population P. Whichever individual with the highest fitness value yields the most effective global solution.

**F. Fitness evaluation**. In order to evaluate a group of individuals, the fitness evaluation is utilized. An individual's level of fitness is the outcome of this algorithm.

These procedures, apart from the initial setup, continue indefinitely until the pausing requirements have been met. A collection of BSA processes is shown in Figure 4. During initialization, the population is seeded with a random population of individuals (P). Individual historical populations ($P^{old}$) are also generated in the Selection-I process. There's a random rearrangement of $P^{old}$ members in this stage, and the $P^{old}$ value is modified. There is no change in $P^{old}$ value after several repetitions. With the help of $P^{old}$ and P, a mutation process develops a new experimental population (mutant). Furthermore, P and Mutant are used to create a population of the final cross-trial (T). P is eventually updated with all of the individuals in T in Selection-II by determining which one has the best fitness and then selecting that individual [48,49]. Those who contributed to the work, as long as you do not meet the halting conditions, you will keep switching between the first and second options. Figure 4 depicts the BSA flowchart in all of its detail.

*3.3. Landslide Inventory Map (LIM) and Landslide Conditioning Factors*

Applying the Artificial Intelligence optimization algorithms (BBO and BSA) in assessing and predicting high-risk regions requires a solid database containing information and factors that influence landslide occurrence [40]. Based on prior research and experts' opinions, 16 parameters influencing slope movements were subsequently identified [50]. Using the DEM layer in ArcGIS 10.3, the slope, gradient, orientation, transverse, and longitudinal curvature layers were generated [51]. Additionally, the Normalized Difference Vegetation Index (NDVI) layer was developed on the USGS website using Landsat satellite images from ETM sensors using Arc GIS 10.3 on 8 January 2019. In each of the one hundred cities evaluated using the geodatabase, the lithological strata and radius from the fracture were derived using the same method. Likewise, lithological layers [52] and fault distance were extracted from the geological maps of thousand cities investigated [41].

Interpolation Distance Weight (IDW) was employed to construct the precipitation layers founded on long-term annual mean rainfall at rain gauge stations in study cities. Thus, (1) distance from the fault, (2) profile curvature, (3) elevation, (4) NDVI, (5) topographic wetness index (TWI), (6) rainfall [53], (7) stream power index (SPI), (8) terrain roughness index (TRI), (9) slope degree, (10) plan curvature, (11) sediment transport index (STI), (12) distance from the river, and (13) distance from the road. Table 1 lists the characteristics that determine the categorization of landslides. Figure 5 demonstrates that ArcGIS 10.3, SPSS 20, and MATLAB were used to assess and execute models and methods. As shown in Figure 6, databases on the causes of landslides in Sanandaj, Saghez, Baneh, Sarvabad, Marivan, and Saghez were developed by extracting data from these layers.

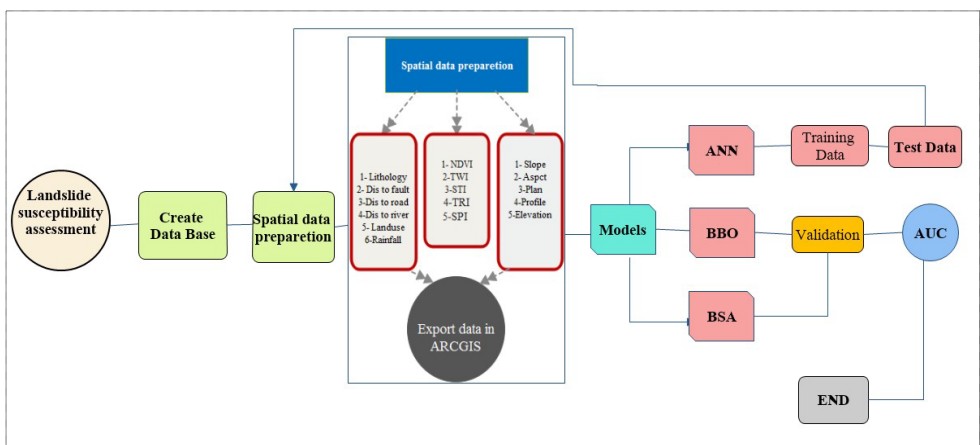

**Figure 5.** Methodological process of the applied model for measuring susceptibility landslides assessment.

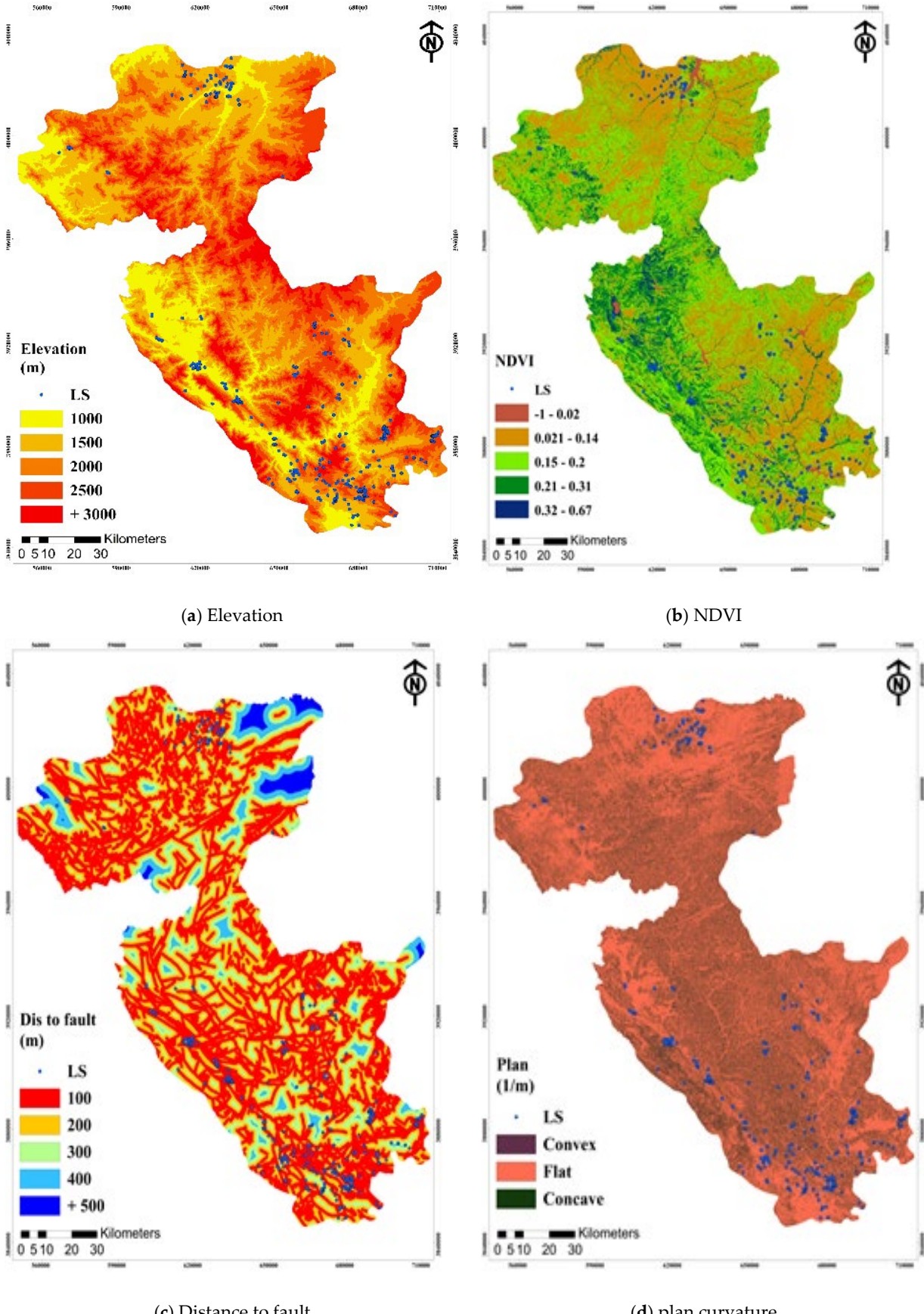

(**a**) Elevation

(**b**) NDVI

(**c**) Distance to fault

(**d**) plan curvature

**Figure 6.** *Cont.*

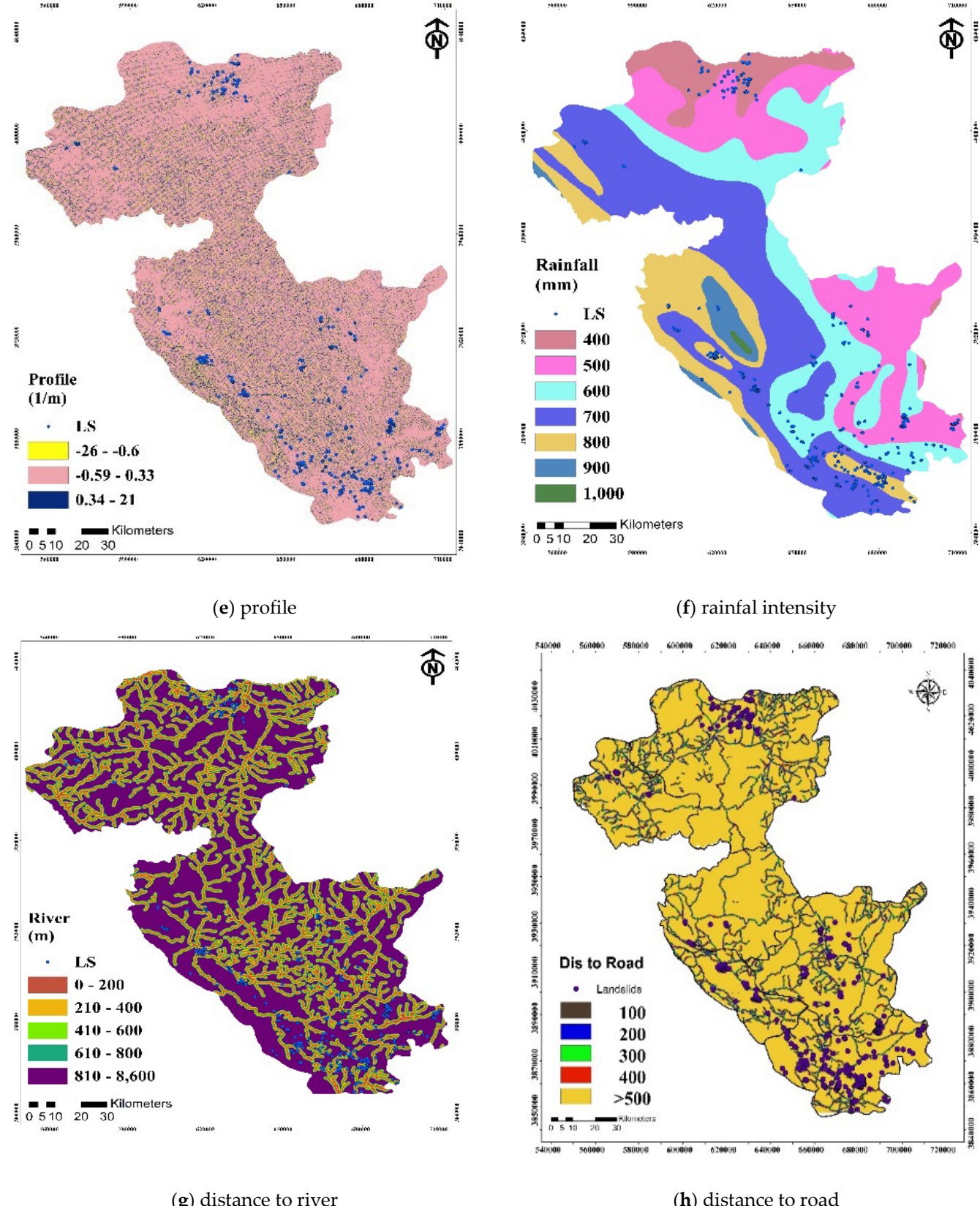

(**e**) profile

(**f**) rainfal intensity

(**g**) distance to river

(**h**) distance to road

**Figure 6.** *Cont.*

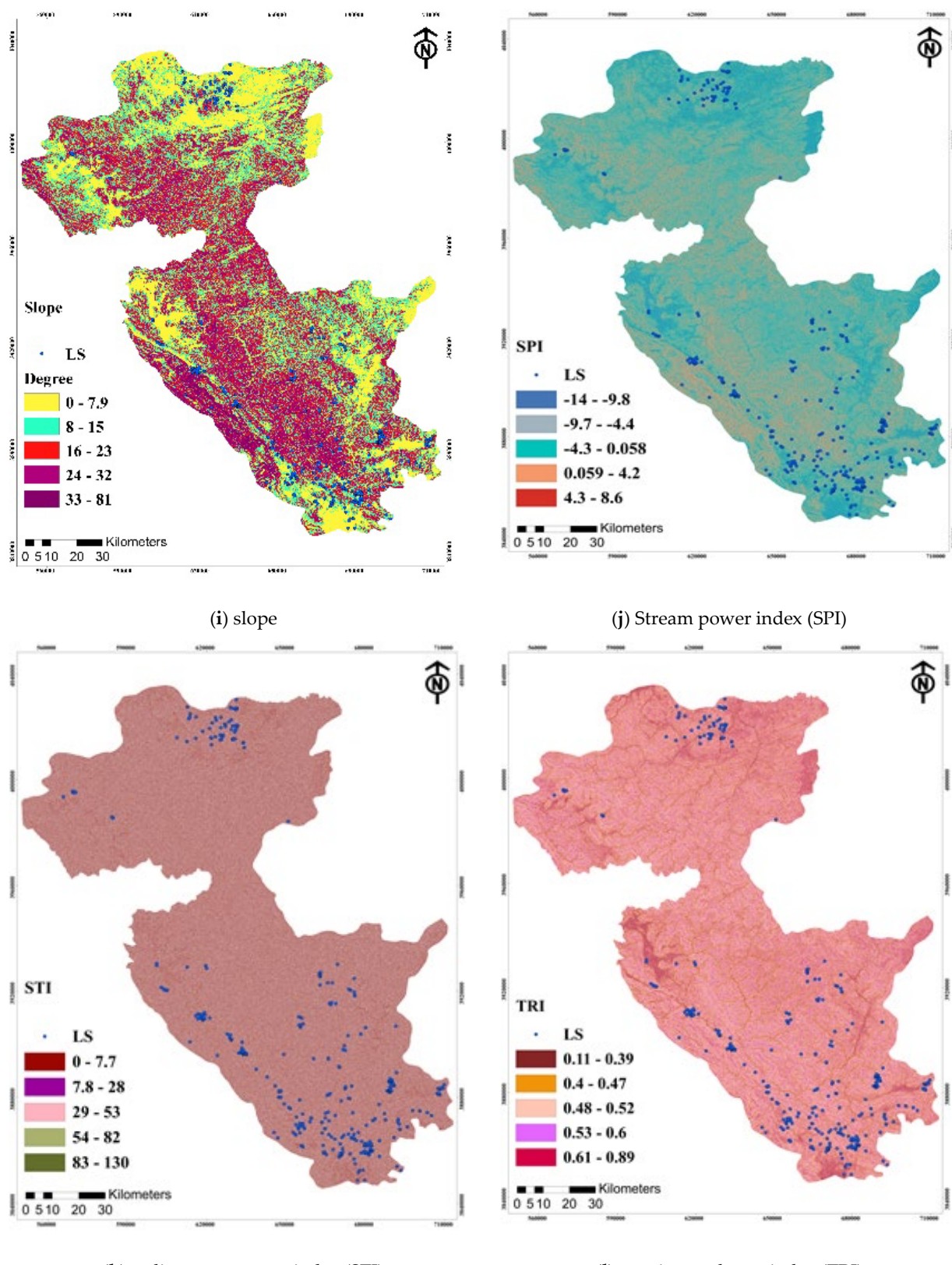

(**i**) slope

(**j**) Stream power index (SPI)

(**k**) sediment transport index (STI)

(**l**) terrain roughness index (TRI)

**Figure 6.** *Cont.*

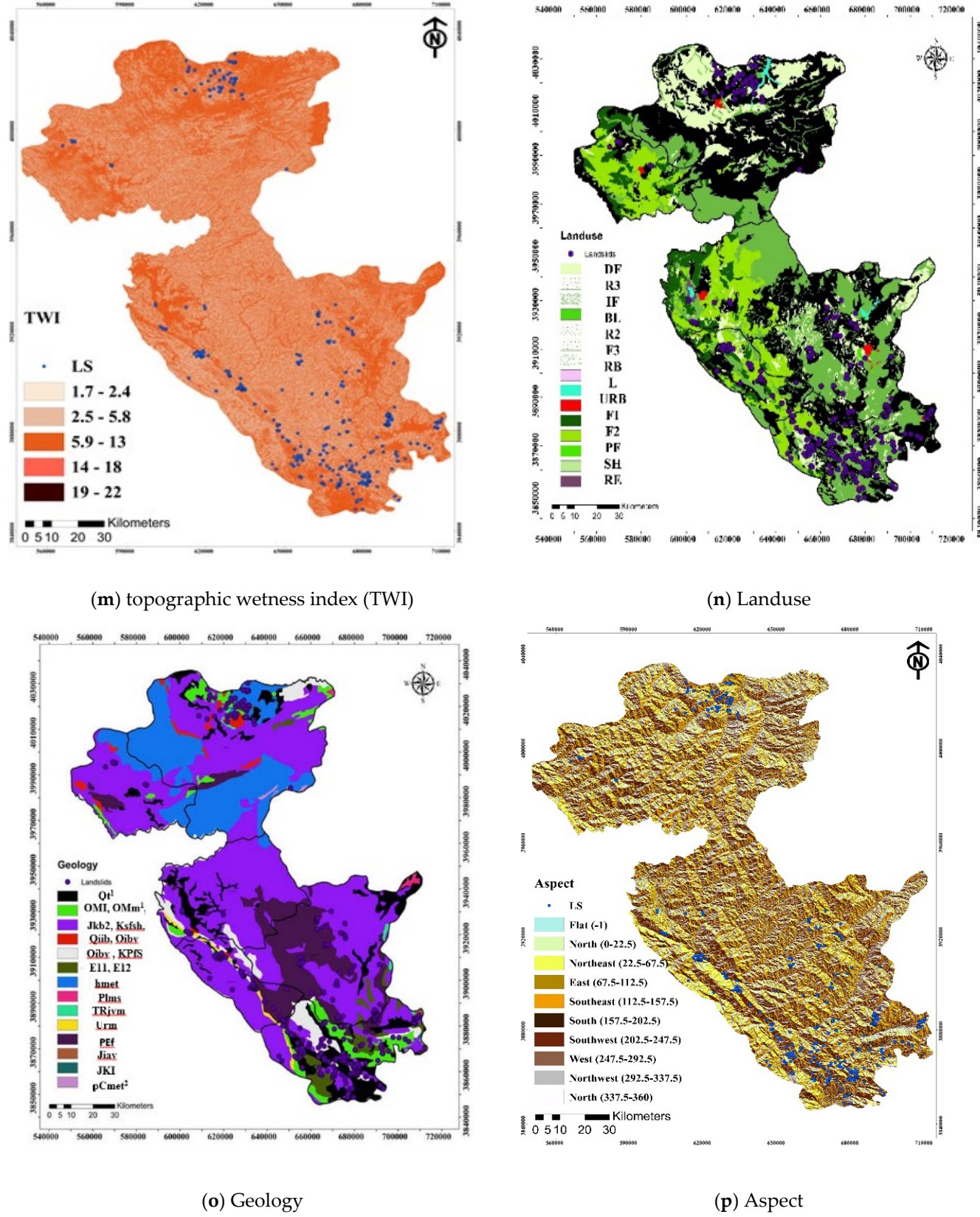

(**m**) topographic wetness index (TWI)                                                        (**n**) Landuse

(**o**) Geology                                                                                                      (**p**) Aspect

**Figure 6.** Maps of landslide conditioning factors in the study area.

**Table 1.** Landslide conditioning factors and their class.

| Factor | Classes | | GIS Data Type | Scale | Classification Method |
|---|---|---|---|---|---|
| Profile | −25.21 | | GRID | 30 m × 30 m | Natural breaks |
| | −1.11 | | | | |
| | 0.34–21 | | | | |
| Plane | Convex | −16.58 | GRID | 30 m × 30 m | Natural breaks |
| | Flat | −0.62 | | | |
| | Concave | 0.22–15 | | | |
| SPI (Stream Power Index) | −8.4 | | GRID | 30 m × 30 m | Manual |
| | −4.3 | | | | |
| | −1.37 | | | | |
| | 0.28–2.2 | | | | |
| | 2.3–8.5 | | | | |
| TWI (Topographic Wetness Index) | 1.7–5.3 | | GRID | 30 m × 30 m | Natural breaks |
| | 5.4–6.7 | | | | |
| | 6.8–8.4 | | | | |
| | 8.5–11 | | | | |
| | 20-Dec | | | | |
| Distance to River | 200 | | Line | 30 m × 30 m | Natural breaks |
| | 400 | | | | |
| | 600 | | | | |
| | 800 | | | | |
| | >800 | | | | |
| Rainfall | 400 | | GRID | 30 m × 30 m | Natural breaks |
| | 500 | | | | |
| | 600 | | | | |
| | 700 | | | | |
| | 800 | | | | |
| NDVI (Normalized Difference Vegetation Index) | −1.04 | | GRID | 30 m × 30 m | Natural breaks |
| | 0.041–0.13 | | | | |
| | 0.17–0.20 | | | | |
| | 0.24–0.32 | | | | |
| | 0.33–0.65 | | | | |
| Slope | <7 | | GRID | 30 m × 30 m | Manual |
| | 15 | | | | |
| | 22 | | | | |
| | 32 | | | | |
| | >80 | | | | |

**Table 1.** *Cont.*

| Factor | Classes | GIS Data Type | Scale | Classification Method |
|---|---|---|---|---|
| Aspect | a-Northwest, b-South, c-North, d-Southeast, e-East, f-West, g-Southwest, e-Northeast, j-North k-Flat, | GRID | 30 m × 30 m | Azimuth classification |
| Land use | Sliding influential factors | Polygon | 1:25,000 | Natural breaks |
| Geology | Qt[1] | Polygon | 1:100,100 | Natural breaks |
|  | OMl, OMm[1], OMl, |  |  |  |
|  | Jkb2, Ksfsh, kussh, K1m, |  |  |  |
|  | Pr |  |  |  |
|  | Qiib, Oibv |  |  |  |
|  | hmet |  |  |  |
|  | Urm |  |  |  |
|  | PEf |  |  |  |
| TRI (Terrain Ruggedness Index) | 0.11–0.38 | GRID | 30 m × 30 m | Natural breaks |
|  | 0.39–0.46 |  |  |  |
|  | 0.47–0.52 |  |  |  |
|  | 0.53–0.6 |  |  |  |
|  | 0.61–0.89 |  |  |  |
| STI (Sediment transport index) | 0–0.45 | GRID | 30 m × 30 m | Natural breaks |
|  | 0.45–7.44 |  |  |  |
|  | 7.45–28.2 |  |  |  |
|  | 28.3–52.7 |  |  |  |
|  | 52.8–82.6 |  |  |  |
| Distance from Road | 100 | Line | 1:25,000 | Manual |
|  | 200 |  |  |  |
|  | 300 |  |  |  |
|  | 400 |  |  |  |
|  | >500 |  |  |  |
| Distance from Fault | 100 | Line | 1:100,100 | Manual |
|  | 200 |  |  |  |
|  | 300 |  |  |  |
|  | 400 |  |  |  |
|  | >500 |  |  |  |
| Elevation | <1000 | GRID | 30 m × 30 m | Natural breaks |
|  | 1500 |  |  |  |
|  | 2000 |  |  |  |
|  | 2500 |  |  |  |
|  | >3000 |  |  |  |

## 4. Results and Discussion

The MATLAB environment tests analyze and simulate model architectures in the study. To find the best design, several networks with different numbers of layers and types of neurons have been built. When the number of layers and neurons in a standard ANN is changed, the accuracy of the models also changes (see Table 2). On average, based on the RMSE and $R^2$ indicators, the best network was created using a feed-forward back-propagation approach with six hidden units (i.e., the *tansig* function and six neurons in the hidden layer) (Table 2). Initial optimization results are used as a starting phase for different optimization methods. The best predictive network results from the model with the highest score (or least rank in Table 2). It is noteworthy that the scores came directly from the model prediction result accuracies. For instance, the lowest RMSE obtained results in a higher score for the specified model. However, for the $R^2$, the higher $R^2$ will result in a higher score. Therefore, the next sections make use of the results of these networks. Figures 7 and 8 further show how the MSE changes when the amount of each neuron per hidden layer increases or decreases. Table 2: A sensitivity study of forecasting landslide susceptibility mapping's number on varying numbers of neurons.

**Table 2.** Change in neuronal density as a predictor for landslide susceptibility mapping: sensitivity analysis.

| Model ID | Number of Neurons | RMSE Training | RMSE Testing | RMSE Total | Train | Scoring Test | Total Data | Total Score | RANK |
|---|---|---|---|---|---|---|---|---|---|
| ANN_1 | 1 | 1.214 | 1.230 | 1.222 | 1 | 1 | 1 | 3 | 10 |
| ANN_2 | 2 | 0.873 | 0.891 | 0.875 | 2 | 2 | 2 | 6 | 9 |
| ANN_3 | 3 | 0.629 | 0.655 | 0.635 | 5 | 4 | 5 | 14 | 6 |
| ANN_4 | 4 | 0.649 | 0.624 | 0.641 | 4 | 5 | 4 | 13 | 7 |
| ANN_5 | 5 | 0.560 | 0.553 | 0.561 | 8 | 9 | 8 | 25 | 3 |
| ANN_6 | 6 | 0.721 | 0.717 | 0.722 | 3 | 3 | 3 | 9 | 8 |
| ANN_7 | 7 | 0.570 | 0.573 | 0.579 | 7 | 7 | 7 | 21 | 4 |
| ANN_8 | 8 | 0.554 | 0.557 | 0.550 | 10 | 10 | 10 | 30 | 1 |
| ANN_9 | 9 | 0.549 | 0.551 | 0.555 | 9 | 8 | 9 | 26 | 2 |
| ANN_10 | 10 | 0.582 | 0.578 | 0.584 | 6 | 6 | 6 | 18 | 5 |

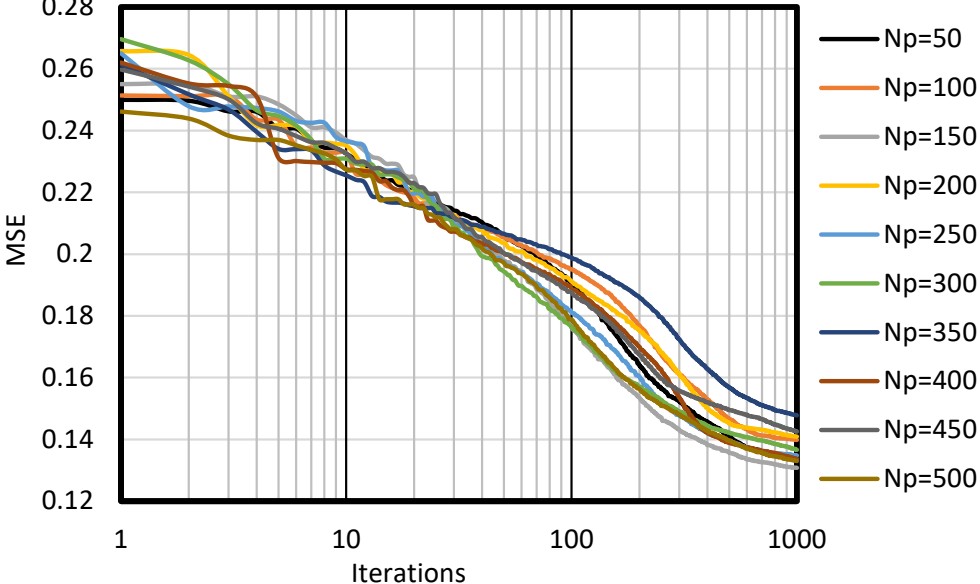

**Figure 7.** The variation of mean squared error versus iterations obtained from the proposed BBOMLP structures in predicting landslide susceptibility mapping.

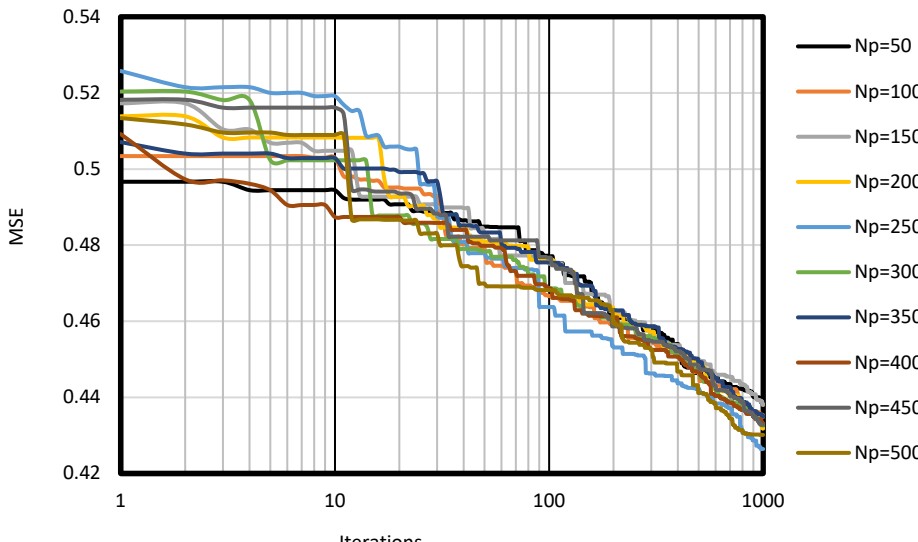

**Figure 8.** The variation of mean squared error as a function of iterations was acquired as from suggested BSAMLP architectures for forecasting flood vulnerability mapping.

The succeeding optimization strategies will be built upon the foundation of the first optimization discovery phase. Thus, these networks' outputs are utilized in the next parts. Better prediction accuracy is seen in structures with less MSE. Regression and classification results may be solved more accurately using the proposed model's estimated values. The proposed BSAMLP architectures in forecasting landslide susceptibility mapping for various hybrid BBOMLP and BSAMLP structures display variation in mean squared error (MSE) against iterations in Figures 6 and 7. The BBO and the BSA have settled on 400 and 450 ($N_{swarm}$) as the optimal solution based on these data.

*Error Analysis*

The findings of the second stage are presented by comparing real data with the anticipated values of the hybrid design. In the majority of instances, receiver operating characteristic (ROC) curves are used to determine the optimal hybrid design (AKA ROC curves). As stated before, the graph depicts how the diagnostic capabilities of a binary classifier system vary when the discriminating threshold is altered. An AUC of one would be desirable, whereas a result of zero would indicate no correlation. There is no connection between a predicted value and its actual zero value. The AUC summarizes ROC curves by assessing a classifier's ability to differentiate across classes. With increasing AUC, the model's ability to differentiate between positive and negative classifications grows. Figures 9 and 10 show the ROC curves for the specified hybrid BBO-MLP and BSA-MLP models. The best prediction model (based on the proposed hybrid BBO and BSA models) was built for population sizes 450 and 400 based on the results of the iteration phase. This outcome is the result of 20,000 MSE modeling and assessment iterations.

Step two of the grading process involves the use of AUC to determine the optimal hybrid designs. In the BBOMLP training datasets, the expected AUC accuracy indices for population sizes of 150, 400, 500, 50, 250, 300, 100, 450, 100, and 350 were 0.914, 0.909, 0.906, 0.906, 0.905, 0.903, 0.899, 0.896, 0.896, and 0.883, respectively (Table 3). Likewise, the BBO-MLP testing datasets generated AUC values of 0.842, 0.834, 0.809, 0.804, 0.801, 0.798, 0.792, 0.788, 0.773, and 0.743 for swarm populations of 150, 300, 500, 200, 250, 450, 400, 100, 50, and 350, respectively. Regarding testing and training predictive modeling outputs, the optimal hybrid technique to forecast landslide susceptibility mapping (e.g., how well the algorithm may forecast landslide susceptibility) has a swarm population of 150. Moreover, it proves that the outcome of phase one closely mirrors those of step two. In the event of BSAMLP (Table 4), the testing and training AUC values were (0.813, 0.805, 0.804, 0.803, 0.799,

0.799, 0.794, 0.795, 0.789, and 0.789) and (0.771, 0.771, 0.771, 0.758, 0.756, 0.727, 0.715, 0.710, 0.706, 0.702, and 0.694) under the same swarm population conditions (Figures 11 and 12).

**Table 3.** The Results of AUC for different BBOMLP proposed structure in predicting the landslide susceptibility mapping.

| Population Size | Network AUC Results | | Scoring | | Total Score | RANK |
|---|---|---|---|---|---|---|
| | Training | Testing | Training | Testing | | |
| 50 | 0.906 | 0.773 | 8 | 2 | 10 | 6 |
| 100 | 0.899 | 0.788 | 4 | 3 | 7 | 9 |
| 150 | 0.914 | 0.842 | 10 | 10 | 20 | 1 |
| 200 | 0.896 | 0.804 | 2 | 7 | 9 | 7 |
| 250 | 0.905 | 0.801 | 6 | 6 | 12 | 5 |
| 300 | 0.903 | 0.834 | 5 | 9 | 14 | 3 |
| 350 | 0.883 | 0.743 | 1 | 1 | 2 | 10 |
| 400 | 0.909 | 0.792 | 9 | 4 | 13 | 4 |
| 450 | 0.896 | 0.798 | 3 | 5 | 8 | 8 |
| 500 | 0.906 | 0.809 | 7 | 8 | 15 | 2 |

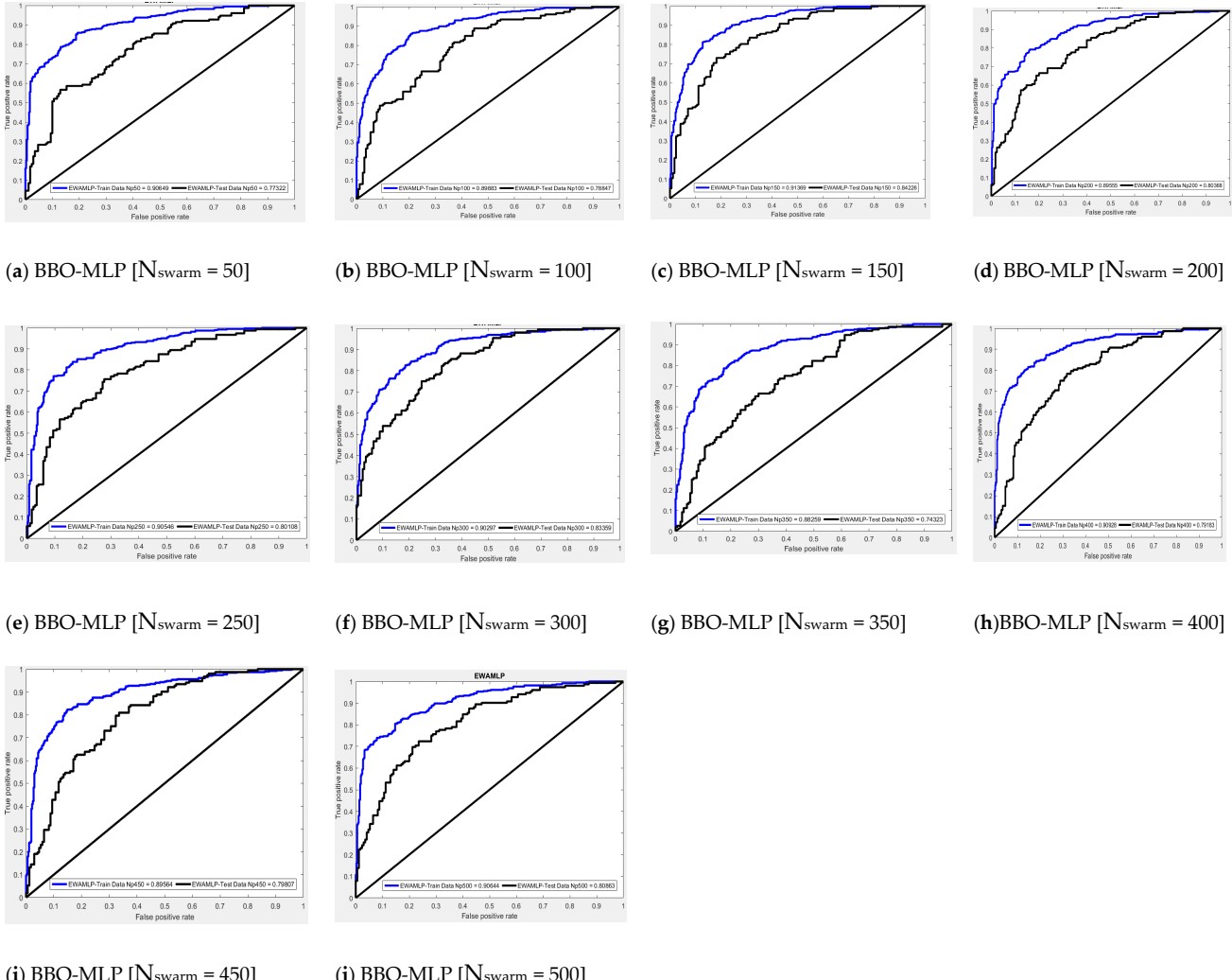

(**a**) BBO-MLP [N$_{swarm}$ = 50]  (**b**) BBO-MLP [N$_{swarm}$ = 100]  (**c**) BBO-MLP [N$_{swarm}$ = 150]  (**d**) BBO-MLP [N$_{swarm}$ = 200]

(**e**) BBO-MLP [N$_{swarm}$ = 250]  (**f**) BBO-MLP [N$_{swarm}$ = 300]  (**g**) BBO-MLP [N$_{swarm}$ = 350]  (**h**)BBO-MLP [N$_{swarm}$ = 400]

(**i**) BBO-MLP [N$_{swarm}$ = 450]  (**j**) BBO-MLP [N$_{swarm}$ = 500]

**Figure 9.** The ROC curves plotted for the (i) training and (ii) testing datasets in BBOMLP algorithm.

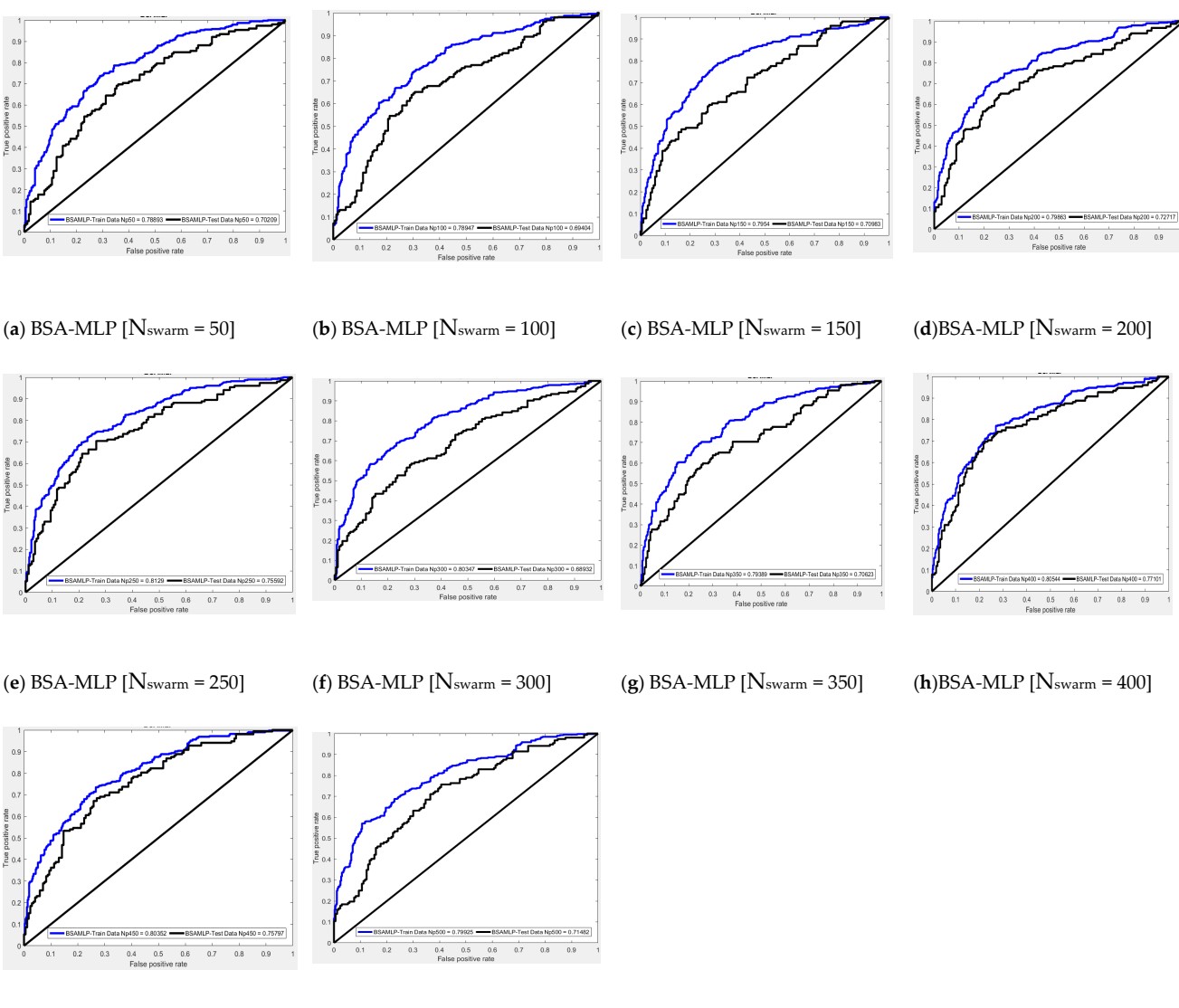

**Figure 10.** The ROC curves plotted for the (i) training and (ii) testing datasets in BSAMLP algorithm.

**Table 4.** The Results of AUC for different BSAMLP proposed structure in predicting the landslide susceptibility mapping.

| Population Size | Network AUC Results | | Scoring | | Total Score | RANK |
|---|---|---|---|---|---|---|
| | Training | Testing | Training | Testing | | |
| 50 | 0.789 | 0.702 | 1 | 3 | 4 | 9 |
| 100 | 0.789 | 0.694 | 2 | 2 | 4 | 9 |
| 150 | 0.795 | 0.710 | 4 | 5 | 9 | 6 |
| 200 | 0.799 | 0.727 | 5 | 7 | 12 | 4 |
| 250 | 0.813 | 0.756 | 10 | 8 | 18 | 2 |
| 300 | 0.803 | 0.689 | 7 | 1 | 8 | 7 |
| 350 | 0.794 | 0.706 | 3 | 4 | 7 | 8 |
| 400 | 0.805 | 0.771 | 9 | 10 | 19 | 1 |
| 450 | 0.804 | 0.758 | 8 | 9 | 17 | 3 |
| 500 | 0.799 | 0.715 | 6 | 6 | 12 | 4 |

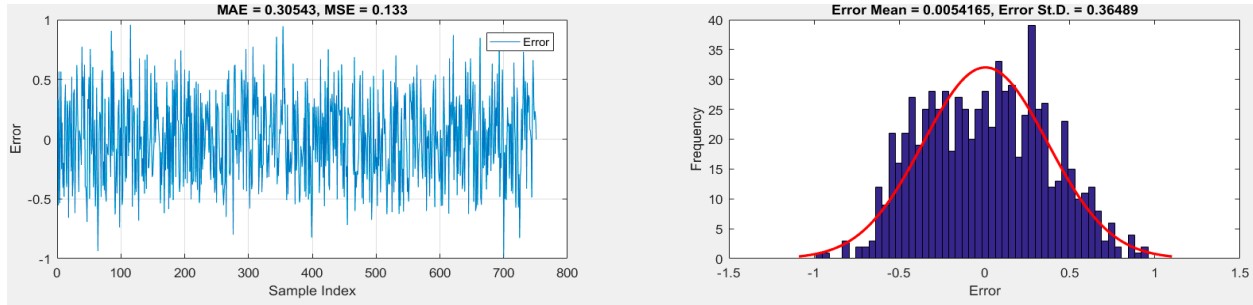

(**a**) Training analysis for the N$_{swarm}$ = 500

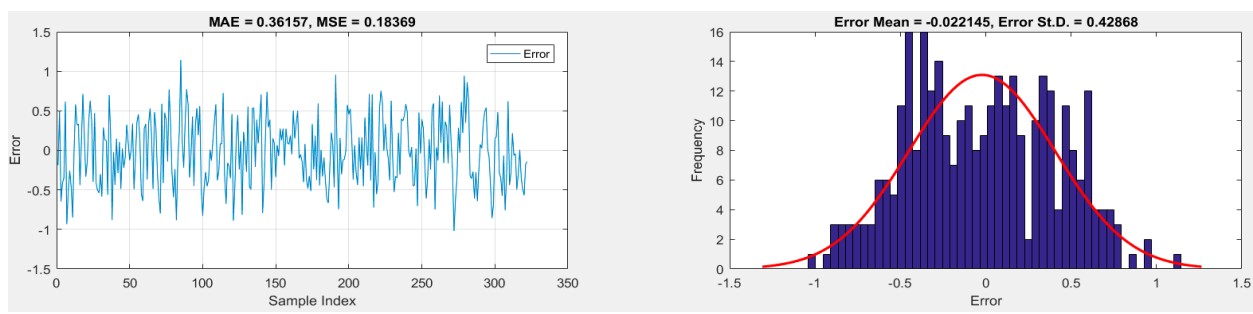

(**b**) Testing analysis for the N$_{swarm}$ = 500

**Figure 11.** BBO-MLP Training and testing analysis.

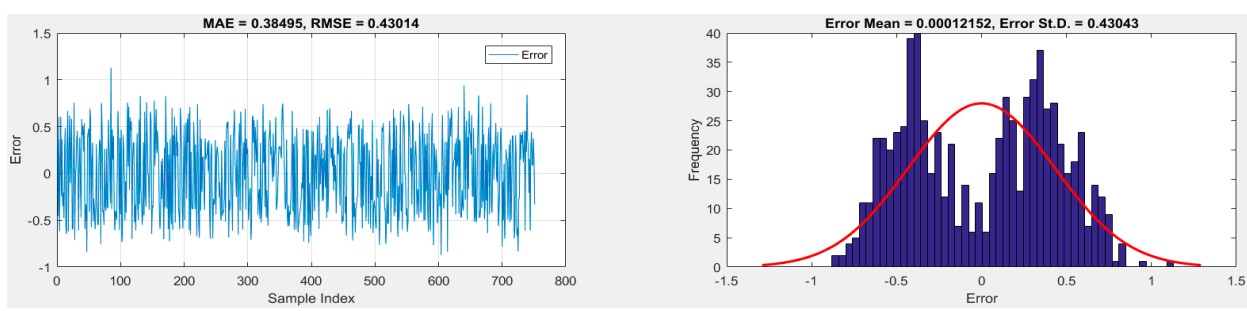

(**a**) Training analysis for the N$_{swarm}$ = 500

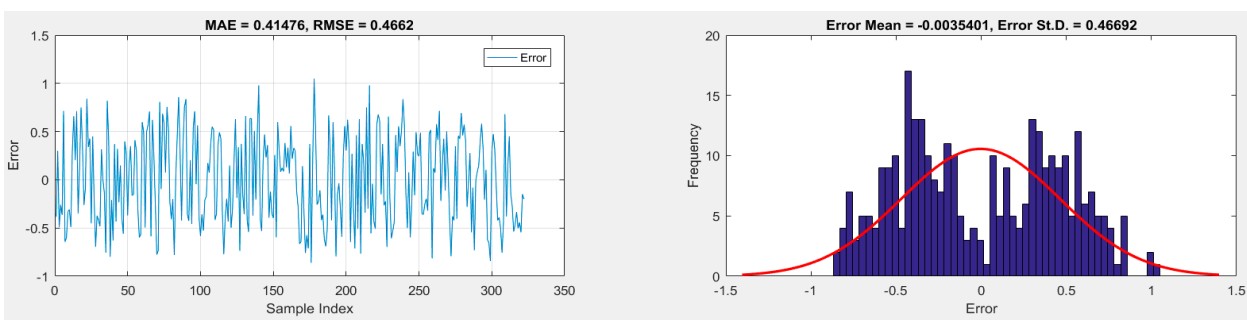

(**b**) Testing analysis for the N$_{swarm}$ = 500

**Figure 12.** BSA-MLP Training and testing analysis.

The final landslide susceptibility mapping using both BBO-MLP and BSA-MLP is shown in Figure 13. This map was prepared based on swarm sizes 150 and 400, which have the highest accuracy and the lowest MSE in the two models. The results show that the BBO-MLP has high accuracy in optimizing. Additionally, according to findings [54], it must

be emphasized that the BBO and BSA has certain appealing qualities as well. Their findings are comparable to the results of this study which have been done on an immense scale and expresses the high accuracy of the algorithm in ANN model optimization operations. Additionally, another Iran study used ANN for susceptibility to landslides [13]. A comparison of its findings with the results of this study has shown that the optimized algorithm ANN had a higher AUC. Prior studies are supplemented by using the BBO-MLP and BSA-MLP models in landslide susceptibility mapping. To some extent, the investigation results may also serve as a source of information.

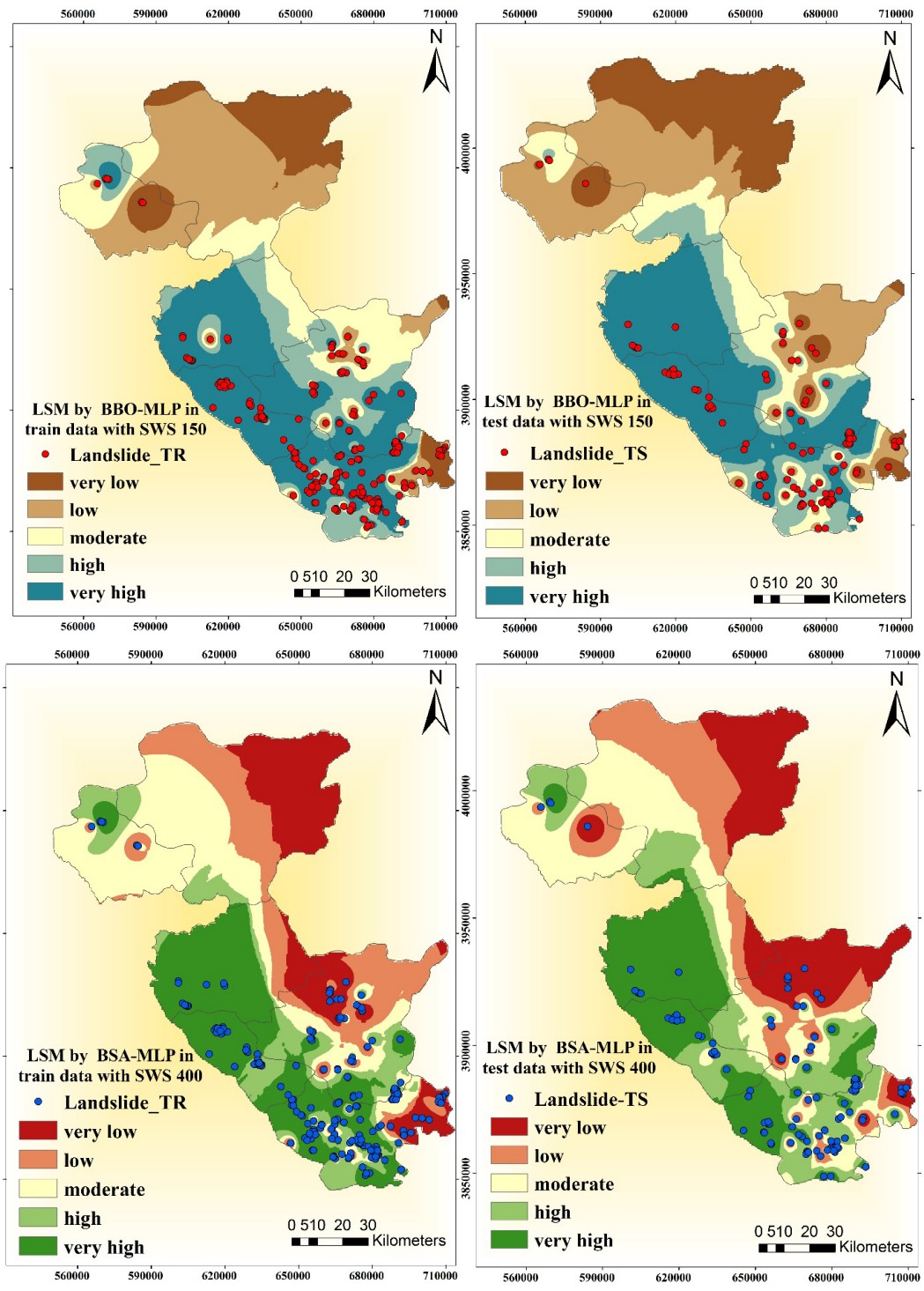

**Figure 13.** The final landslide susceptibility mapping using both BBOMLP and BSAMLP.

Additionally, the maps indicate that the northern part of the study area in western Iran has fewer landslides and its sensitivity to landslides is low. However, the northwest and southeast part of the study area in Kurdistan province is in accordance with mountainous areas and has a high potential for the risks of landslides, especially in the rainy season. As can be seen in the map, circular points show the landslide points in the status quo, which adapt to high-risk and high-risk classes indicating the high accuracy of the two algorithms used for optimization. Other areas that are classified in the northern part of the moderate risk indicate that according to the model forecasts these areas also have the potential for landslides and should be taken into account in crisis management planning.

## 5. Conclusions

Hybrid algorithms are becoming more popular as a solution to a variety of complex issues. Two new methods, BBO and BSA, have improved landslide susceptibility mapping. For identifying the spatial link between landslide-conditioning elements, an MLP neural network was used. Conditioning factors were evaluated based on factors, including; elevation, slope aspect, slope angle, NDVI, distance to fault, plan curvature, profile curvature, rainfall, distance from the river, distance to road, SPI, STI, TRI, TWI, land use, and geology. There were 1072 sites in the landslide inventory database, separated into two groups for training and testing the model, each comprising of 536 landslides. The results showed AUC values; the Optimized metaheuristic algorithm was calculated to be 0.842 in swarm size 150 for BBO-MLP and BSA-MLP, it was obtained in swarm size 400. The landslide susceptibility map was created based on the best-fit hybrid model specified as an optimized category with a higher AUC. The real inspiration for using BBO and BSA algorithms in the current study was the widespread use of conventional optimization for landslide susceptibility mapping. The algorithm's computational parameters were effectively optimized via a synthetic neural network. That model's ideal structure was ultimately discovered after significant trial and error. There are fourteen landslide conditioning factors in the spatial database in different ecological, geographical, and structural dimensions. By random selection, the proposed models are trained on seventy percent of the identified landslides using an accidental sampling method, and their precision is evaluated on the residual thirty percent. To evaluate the accuracy of the forecasting models, the area under the curve (AUC) criterion was used. The precision indicator of the area underneath the receiving operating characteristic curve (AUROC) demonstrated that the maps created by the BBO-ANN (with an AUROC value of 0.842) are more precise compared to those generated by the BSA-ANN (with an AUROC value of 0.771). Additionally, the corresponding estimated AUCs in this regard were for BBO-MLP 0.842, 0.834, 0.809, 0.804, 0.801, 0.798, 0.792, 0.788, 0.773, and 0.743 and BSA-MLP 0.813, 0.805, 0.804, 0.803, 0.799, 0.799, 0.794, 0.795, 0.789, and 0.789, respectively. A 150-person swarm size characterizes the best-fit hybrid model for predicting landslide susceptibility mapping, and it belongs to the BBO-MLP model. According to the findings, these algorithms functioned well to enhance the MLP's learning potential. Additionally, adding metaheuristic algorithms such as BBO may significantly enhance the performance of the ANN with a drop in prediction MSE of 1.230 to 0.551 percent. Three well-known accuracy criteria—MSE, RMSE, and AUROC—were employed to create a rating system that compared the practicality of the utilized model. The result of this part showed that effectively the BBO and BSA algorithms increase the MLP's capacity for learning.

Therefore, Evolutionary science can be utilized as a first step to improve the reliability of neural computing. The BBO-MLP ensemble takes less time to train correctly than the BSA-MLP, according to the data. Additionally, the BBO algorithm enhances the accuracy of the MLP. Moreover, referring to the calculated total ranking scores of 10, 7, and 5, it becomes apparent that the BBO performs more efficiently than the BSA in optimizing the MLP. However, the highest prediction accuracy is found in structures with less error and in this research, each of the two algorithms achieved a lower error ($RMSE_{\text{BBO-MLP and BSA-MLP}} = 0.551$ and 0.557). Additionally, despite the superiority of the BBO-MLP in learning landslide

patterns, both ensembles presented a close prediction accuracy ($AUC_{BBO-MLP} = 0.842$ and $AUC_{BSA-MLP} = 0.771$). In general, this BBO-MLP is more effective in improving neural network performance in this paper.

**Author Contributions:** Conceptualization, methodology, writing—original draft preparation, H.M. and P.J.C.; investigation, visualization, writing—original draft preparation, A.A.D.; visualization, formal analysis, M.A.C. and M.S.; writing—review and editing, H.M. and B.N.L.; H.M. and B.N.L. All authors have read and agreed to the published version of the manuscript.

**Funding:** This research received no external funding.

**Institutional Review Board Statement:** All authors have participated in (a) conception and design, or analysis and interpretation of the data; (b) drafting the article or revising it critically for important intellectual content; and (c) approval of the final version. This manuscript has not been submitted to, nor is under review at, another journal or other publishing venue. The authors have no affiliation with any organization with a direct or indirect financial interest in the subject matter discussed in the manuscript.

**Informed Consent Statement:** Informed consent was obtained from all subjects involved in the study.

**Data Availability Statement:** The data that support the findings of this study are available from the corresponding author upon reasonable request.

**Acknowledgments:** We would like to express our appreciation to all the participants, without whom this study would be impossible.

**Conflicts of Interest:** The authors declare no conflict of interest.

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
