# Peer review of "Multilayer Perceptron and Their Comparison with Two Nature-Inspired Hybrid Techniques of Biogeography-Based Optimization (BBO) and Backtracking Search Algorithm (BSA) for Assessment of Landslide Susceptibility"

_land, doi:10.3390/land12010242_

Round 1
Reviewer 1 Report
This paper focused on the landslide susceptibility zoning with Machine learning, which is a hot topic in nowadays. The authors select some factors of the geo-environments with new idea, like topographic wetness index, sediment transport index. Anyway, the authors emphsis on the optimization of ML algorithm, and did not give why choose these factors and its effect.
Although the machine learning is popular, it still can not give good explaination of some algorithm. The BBO and BSA is some kind a ecology model, part of this model related with its inner species' selection, maybe different to the non-life rock and soil. Perhaps there are some work can be done here.
As the authors mentioned, the swarm population affected the training effect, and this population is lack of explaination, why the swarm size 150 and 400 are the best performance? The final suceptibility level is also different with traditional one, they are four map and even some isolated point is obvious. the result need more explain or adjustment.
Author Response
Title: Multilayer perceptron and their comparison with two nature-inspired
hybrid techniques of biogeography-based optimization (BBO) and backtracking
search algorithm (BSA) for assessment of Landslide susceptibility”
Journal: Land
Manuscript ID: land-2112773
Responses to the comments
|
|
Issue |
Authors’ answer |
Revisions |
|
|
|
Many thanks for the valuable and constructive comments of the reviewers. An attempt was made to accurately apply all comments to the text of the article. Revisions made to the text of the article include:
|
|
|
|
|
Responses to the First Reviewer |
|
|
1 |
This paper focused on the landslide susceptibility zoning with Machine learning, which is a hot topic in nowadays. The authors select some factors of the geo-environments with new idea, like topographic wetness index, sediment transport index. Anyway, the authors emphsis on the optimization of ML algorithm, and did not give why choose these factors and its effect. Although the machine learning is popular, it still can not give good explaination of some algorithm. The BBO and BSA is some kind a ecology model, part of this model related with its inner species' selection, maybe different to the non-life rock and soil. Perhaps there are some work can be done here. |
Thanks for your comments. The answer to this comment point to point is presented in the text and in the section as follows:
This paper focused on the landslide susceptibility zoning with Machine learning, which is a hot topic in nowadays. The authors select some factors of the geo-environments with new idea, like topographic wetness index, sediment transport index. Anyway, the authors emphsis on the optimization of ML algorithm, and did not give why choose these factors and its effect.
Many studies have been conducted in the field of landslide susceptibility zoning in different regions of the world. But most of them have ignored these two TWI & STI indicators. These two indicators are especially effective in mountainous areas that have a high potential for landslides. The area studied in this research is also mountainous, so it was necessary to consider these two indicators as well. It should also be mentioned that in landslide zoning studies, the more the number of indicators to identify risky areas, the higher the accuracy of the study. Since the mechanism of occurrence of this hazard in different regions with different geographical conditions on a micro to macro scale; it is different, so the researchers should use the indicators to identify and zone this hazard with the knowledge of the geographical conditions and spatial and social constructions of that place. Therefore, in this study, these two indicators were also considered.
Although the machine learning is popular, it still can not give good explaination of some algorithm. The BBO and BSA is some kind a ecology model, part of this model related with its inner species' selection, maybe different to the non-life rock and soil. Perhaps there are some work can be done here
Today, artificial intelligence techniques have many applications in various sciences, especially biological sciences, engineering, medicine, earth sciences, etc. Although this model is simulated from the human neural network, it is used in various fields in a practical and research way. Neural network algorithms have no restrictions on its application in different fields, in any problem that has a network nature and in which a range of factors are systematically related and have backward/forward relationships and It can be used if it has a role in the occurrence of a problem or risk. This model requires the use of optimization algorithms in order to be used in different fields due to some problems and limitations in it. In fact, BBO & BSO algorithms, by simulating the problem in the real world and converting it into mathematical equations, use the same strategy of living organisms to provide the optimal solution among the existing solutions. In fact, all living organisms use biological and optimal strategies for survival and even in the face of dangers, the mechanism of these algorithms is to convert these biological strategies into mathematical equations to solve the problems of human societies in the face of natural or human dangers. In the case that these strategies are used to solve neural network problems to face natural hazards, optimization algorithms can solve complex problems intelligently and in accordance with mathematical equations, and the accuracy and accuracy of neural network operations in identifying and increasing risk assessment.
|
Please see the whole manuscript |
|
2 |
As the authors mentioned, the swarm population affected the training effect, and this population is lack of explaination, why the swarm size 150 and 400 are the best performance? The final suceptibility level is also different with traditional one, they are four map and even some isolated point is obvious. the result need more explain or adjustment. |
Thanks to the valuable comments of the reviewer, the result was corrected and revised in the text of the article were made as follows:
Conditioning factors were evaluated based on factors, including; elevation, slope aspect, slope angle, NDVI, distance to fault, plan curvature, profile curvature, rainfall, distance from the river, distance to road, SPI, STI, TRI, TWI, land use, and geology. There were 1072 sites in the landslide inventory database, separated into two groups for training and testing the model, each comprising 536 landslides. The results showed AUC values; the Optimized metaheuristic algorithm was calculated to be 0.842 in swarm size 150 for BBO-MLP and BSA-MLP, it was obtained in swarm size 400. The landslide susceptibility map was created based on the best-fit hybrid model specified as an optimized category with a higher AUC. The real inspiration for using BBO & BSA algorithms in the current study was the widespread use of conventional optimization for landslide susceptibility mapping. The algorithm's computational parameters were effectively optimized via a synthetic neural network. That model's ideal structure was ultimately discovered after significant trial and error. There are fourteen landslide conditioning factors in the spatial database in different ecological, geographical, and structural dimensions. By random selection, the proposed models are trained on seventy percent of the identified landslides using an accidental sampling method, and their precision is evaluated on the residual thirty percent. To evaluate the accuracy of the forecasting models, the area under the curve (AUC) criterion was used. The precision indicator of the area underneath the receiving operating characteristic curve (AUROC) demonstrated that the maps created by the BBO-ANN (with an AUROC value of 0.842) are more precise compared to those generated by the BSA-ANN (with an AUROC value of 0.771). Also, the corresponding estimated AUCs in this regard were for BBO-MLP 0.842, 0.834, 0.809, 0.804, 0.801, 0.798, 0.792, 0.788, 0.773, and 0.743 and BSA-MLP 0.813, 0.805, 0.804, 0.803, 0.799, 0.799, 0.794, 0.795, 0.789, and 0.789, respectively. A 150-person swarm size characterizes the best-fit hybrid model for predicting landslide susceptibility mapping, and it belongs to the BBO-MLP model. According to the findings, these algorithms functioned well to enhance the MLP's learning potential. Additionally, adding metaheuristic algorithms such as BBO may significantly enhance the performance of the ANN with a drop in prediction MSE of 1.230 to 0.551 percent. Three well-known accuracy criteria—MSE, RMSE, and AUROC—were employed to create a rating system that compared the practicality of the utilized model. The result of this part showed that effectively the BBO & BSA algorithms increase the MLP's capacity for learning.
|
Please see page 27 -28 |
In the end, thanks to the respected editor and reviewers who spent a lot of time and energy scrutinizing this manuscript. If the explanations and corrections made are not enough, please do not hesitate to contact us for further improvements.
All the best,
Corresponding author, also on behalf of my co-authors

Reviewer 2 Report
1. In introduction chapter, the content introduced is landslide displacement prediction, which does not match the title.
2. In 3.2 chapter, The format of Siobserved and Sipredicted is different.
3. The introduction and use of the content is confusing. Formula 1 explains MSE, Figure 3 introduces RMSE, MAE, R2 and MSE, and the following text uses RMSE.
4. The formula introduction is not uniform, using sequence number (1) and Equation 2.
5. The formula is not used uniformly, using Equation 1 and Eq. (2).
6. Simulation environment Settings, model parameters need to be described in detail.
7. Lack of detailed comparisons between MLP and BBOMLP and BSAMLP. The ability of BBO and BSA to enhance MLP is not stated.
8. The Scoring system needs to be explained in detail.
Author Response
|
|
Issue |
Authors’ answer |
Revisions |
|||
|
|
|
Many thanks for the valuable and constructive comments of the reviewers. An attempt was made to accurately apply all comments to the text of the article. Revisions made to the text of the article include:
|
|
|||
|
1 |
In introduction chapter, the content introduced is landslide displacement prediction, which does not match the title |
Thank you very much for your comments. It was a typo mistake during the proofreading that used this keyword. We normally use “landslide susceptibility mapping/maps/assessment/analysis/ evaluation/ zonation/ zoning”.
Please kindly be informed that some sections of the introduction are changed and we believe that the revised version is in much better form that the initial submission version.
In this regard, researchers are now using meta-heuristic strategies to improve efficiency due to the limitations of current models, including local minimum and dimension dangers [35]. In this respect, all of these meta-heuristic approaches have a great capacity to resolve optimization issues, and for such reason, they have indeed been implemented in several scientific disciplines. The algorithms have several characteristics, and the majority are population-based techniques. Throughout the calculations, we could perhaps find the best design for each of them. It might be beneficial to create a novel technique that enhances the process or outcomes of optimization. These techniques are used to find high-quality solutions that are based on the best possible computing structure(Tian, Wang, Chen, Zhang, & Qin, 2021). Several advanced strategies (including parallel computation, multi-agent systems, and decomposition of the search space)(Jun Li et al., 2017) are often used in hybrid metaheuristic algorithms. The problems are solved collaboratively by a proactive search agents group acting individually and with parallel computation. They solved many large-scale distributed and dynamic systems with successful results(Reza Naji, Shadravan, Mousa Jafarabadi, & Momeni, 2022). Previous studies have shown that not estimating the participation of each parameter in the classification by the optimized ANN model is one of the primary challenges of neural network model optimization algorithms. It also has some limitations and drawbacks, including high computational power requirements and a significant computation time for determining the final result. In cases where immediate results are required, both weaknesses can be problematic(Chen, Chen, Tsangaratos, Ilia, & Wang, 2020).
|
Please see pages 3-4 |
|||
|
2 |
In 3.2 chapter, The format of Siobserved and Sipredicted is different. |
Thanks much for your comments. I think you mean equation 1. I have double-checked and the format is corrected. Kindly double-check and see if it is correctly revised.
|
Please see page 8 |
|||
|
3 |
The introduction and use of the content is confusing. Formula 1 explains MSE, Figure 3 introduces RMSE, MAE, R2 and MSE, and the following text uses RMSE. |
Thank you very much for your comment. The mentioned comments are corrected based on the respected reviewer's comments.
The below text is used accordingly.
In order to find the minimum error (e.g. finding the best predictive network) during the new model evaluation, we used the term mean square error (here abbreviated as MSE) as the objective function. It indicates the quality of each iteration's resulting solution. In Equation 1, the MSE is given as a formula.
|
Please see page 8 and the whole manuscript |
|||
|
4 |
The formula introduction is not uniform, using sequence number (1) and Equation 2. |
The mathematical expressions and equations were revised in the article. |
Please see pages 8-12 |
|||
|
5 |
The formula is not used uniformly, using Equation 1 and Eq. (2). |
The mathematical expressions and equations were revised in the article. |
Please see pages 8-12 |
|||
|
6 |
Simulation environment Settings, model parameters need to be described in detail. |
Thank you for your comments. We need to first bring your attention to Figure 3 where all the model descriptions are discussed. Similarly Figures 4 and 5 is also described the methods individually. Thereafter in Figure 6, we have shown 16 different data layer inputs that we have employed and that is the most useful spatial distribution basis method that is available in such landslide susceptibility mapping studies. All the input layers are shown and legends are used based on the influential factors that impact the landslide occurrence in the study area. This issue is also well described in section 3.3.
|
Please see page 27 -28 |
|||
|
7 |
Lack of detailed comparisons between MLP and BBOMLP and BSAMLP. The ability of BBO and BSA to enhance MLP is not stated. |
Thanks to the valuable comments of the reviewer, comparisons between MLP and BBOMLP and BSAMLP and ability them for enhancing ANN-MLP revised in the text of the article were made as follows:
Conditioning factors were evaluated based on factors, including; elevation, slope aspect, slope angle, NDVI, distance to fault, plan curvature, profile curvature, rainfall, distance from the river, distance to road, SPI, STI, TRI, TWI, land use, and geology. There were 1072 sites in the landslide inventory database, separated into two groups for training and testing the model, each comprising 536 landslides. The results showed AUC values; the Optimized metaheuristic algorithm was calculated to be 0.842 in swarm size 150 for BBO-MLP and BSA-MLP, it was obtained in swarm size 400. The landslide susceptibility map was created based on the best-fit hybrid model specified as an optimized category with a higher AUC. The real inspiration for using BBO & BSA algorithms in the current study was the widespread use of conventional optimization for landslide susceptibility mapping. The algorithm's computational parameters were effectively optimized via a synthetic neural network. That model's ideal structure was ultimately discovered after significant trial and error. There are fourteen landslide conditioning factors in the spatial database in different ecological, geographical, and structural dimensions. By random selection, the proposed models are trained on seventy percent of the identified landslides using an accidental sampling method, and their precision is evaluated on the residual thirty percent. To evaluate the accuracy of the forecasting models, the area under the curve (AUC) criterion was used. The precision indicator of the area underneath the receiving operating characteristic curve (AUROC) demonstrated that the maps created by the BBO-ANN (with an AUROC value of 0.842) are more precise compared to those generated by the BSA-ANN (with an AUROC value of 0.771). Also, the corresponding estimated AUCs in this regard were for BBO-MLP 0.842, 0.834, 0.809, 0.804, 0.801, 0.798, 0.792, 0.788, 0.773, and 0.743 and BSA-MLP 0.813, 0.805, 0.804, 0.803, 0.799, 0.799, 0.794, 0.795, 0.789, and 0.789, respectively. A 150-person swarm size characterizes the best-fit hybrid model for predicting landslide susceptibility mapping, and it belongs to the BBO-MLP model. According to the findings, these algorithms functioned well to enhance the MLP's learning potential. Additionally, adding metaheuristic algorithms such as BBO may significantly enhance the performance of the ANN with a drop in prediction MSE of 1.230 to 0.551 percent. Three well-known accuracy criteria—MSE, RMSE, and AUROC—were employed to create a rating system that compared the practicality of the utilized model. The result of this part showed that effectively the BBO & BSA algorithms increase the MLP's capacity for learning. |
|
|||
|
8 |
The Scoring system needs to be explained in detail. |
Thank you for your valuable comments. Kindly be noted that below text are added to the section 4 in order to provide more illustrations for the used scoring system. This systems are used in many other literature review too but we also believed that is the best to be discussed here too.
“The best predictive network results from the model with the highest score (or least rank in Table 2). It is noteworthy that, the scores came directly from the model prediction result accuracies. For instance, the lowest RMSE obtained results in a higher score for the specified model. However, for the R2, the higher R2 will result in a higher score. Therefore, the next sections make use of the results of these networks. Figures 7 and 8 further show how the MSE changes when the amount of each neuron per hidden layer increases or decreases. Table 2: A sensitivity study of forecasting landslide susceptibility mapping's number on varying numbers of neurons. |
|
Issue |
Authors’ answer |
Revisions |
|
|
|
Many thanks for the valuable and constructive comments of the reviewers. An attempt was made to accurately apply all comments to the text of the article. Revisions made to the text of the article include:
|
|
|||
|
1 |
In introduction chapter, the content introduced is landslide displacement prediction, which does not match the title |
Thank you very much for your comments. It was a typo mistake during the proofreading that used this keyword. We normally use “landslide susceptibility mapping/maps/assessment/analysis/ evaluation/ zonation/ zoning”.
Please kindly be informed that some sections of the introduction are changed and we believe that the revised version is in much better form that the initial submission version.
In this regard, researchers are now using meta-heuristic strategies to improve efficiency due to the limitations of current models, including local minimum and dimension dangers [35]. In this respect, all of these meta-heuristic approaches have a great capacity to resolve optimization issues, and for such reason, they have indeed been implemented in several scientific disciplines. The algorithms have several characteristics, and the majority are population-based techniques. Throughout the calculations, we could perhaps find the best design for each of them. It might be beneficial to create a novel technique that enhances the process or outcomes of optimization. These techniques are used to find high-quality solutions that are based on the best possible computing structure(Tian, Wang, Chen, Zhang, & Qin, 2021). Several advanced strategies (including parallel computation, multi-agent systems, and decomposition of the search space)(Jun Li et al., 2017) are often used in hybrid metaheuristic algorithms. The problems are solved collaboratively by a proactive search agents group acting individually and with parallel computation. They solved many large-scale distributed and dynamic systems with successful results(Reza Naji, Shadravan, Mousa Jafarabadi, & Momeni, 2022). Previous studies have shown that not estimating the participation of each parameter in the classification by the optimized ANN model is one of the primary challenges of neural network model optimization algorithms. It also has some limitations and drawbacks, including high computational power requirements and a significant computation time for determining the final result. In cases where immediate results are required, both weaknesses can be problematic(Chen, Chen, Tsangaratos, Ilia, & Wang, 2020).
|
Please see pages 3-4 |
|||
|
2 |
In 3.2 chapter, The format of Siobserved and Sipredicted is different. |
Thanks much for your comments. I think you mean equation 1. I have double-checked and the format is corrected. Kindly double-check and see if it is correctly revised.
|
Please see page 8 |
|||
|
3 |
The introduction and use of the content is confusing. Formula 1 explains MSE, Figure 3 introduces RMSE, MAE, R2 and MSE, and the following text uses RMSE. |
Thank you very much for your comment. The mentioned comments are corrected based on the respected reviewer's comments.
The below text is used accordingly.
In order to find the minimum error (e.g. finding the best predictive network) during the new model evaluation, we used the term mean square error (here abbreviated as MSE) as the objective function. It indicates the quality of each iteration's resulting solution. In Equation 1, the MSE is given as a formula.
|
Please see page 8 and the whole manuscript |
|||
|
4 |
The formula introduction is not uniform, using sequence number (1) and Equation 2. |
The mathematical expressions and equations were revised in the article. |
Please see pages 8-12 |
|||
|
5 |
The formula is not used uniformly, using Equation 1 and Eq. (2). |
The mathematical expressions and equations were revised in the article. |
Please see pages 8-12 |
|||
|
6 |
Simulation environment Settings, model parameters need to be described in detail. |
Thank you for your comments. We need to first bring your attention to Figure 3 where all the model descriptions are discussed. Similarly Figures 4 and 5 is also described the methods individually. Thereafter in Figure 6, we have shown 16 different data layer inputs that we have employed and that is the most useful spatial distribution basis method that is available in such landslide susceptibility mapping studies. All the input layers are shown and legends are used based on the influential factors that impact the landslide occurrence in the study area. This issue is also well described in section 3.3.
|
Please see page 27 -28 |
|||
|
7 |
Lack of detailed comparisons between MLP and BBOMLP and BSAMLP. The ability of BBO and BSA to enhance MLP is not stated. |
Thanks to the valuable comments of the reviewer, comparisons between MLP and BBOMLP and BSAMLP and ability them for enhancing ANN-MLP revised in the text of the article were made as follows:
Conditioning factors were evaluated based on factors, including; elevation, slope aspect, slope angle, NDVI, distance to fault, plan curvature, profile curvature, rainfall, distance from the river, distance to road, SPI, STI, TRI, TWI, land use, and geology. There were 1072 sites in the landslide inventory database, separated into two groups for training and testing the model, each comprising 536 landslides. The results showed AUC values; the Optimized metaheuristic algorithm was calculated to be 0.842 in swarm size 150 for BBO-MLP and BSA-MLP, it was obtained in swarm size 400. The landslide susceptibility map was created based on the best-fit hybrid model specified as an optimized category with a higher AUC. The real inspiration for using BBO & BSA algorithms in the current study was the widespread use of conventional optimization for landslide susceptibility mapping. The algorithm's computational parameters were effectively optimized via a synthetic neural network. That model's ideal structure was ultimately discovered after significant trial and error. There are fourteen landslide conditioning factors in the spatial database in different ecological, geographical, and structural dimensions. By random selection, the proposed models are trained on seventy percent of the identified landslides using an accidental sampling method, and their precision is evaluated on the residual thirty percent. To evaluate the accuracy of the forecasting models, the area under the curve (AUC) criterion was used. The precision indicator of the area underneath the receiving operating characteristic curve (AUROC) demonstrated that the maps created by the BBO-ANN (with an AUROC value of 0.842) are more precise compared to those generated by the BSA-ANN (with an AUROC value of 0.771). Also, the corresponding estimated AUCs in this regard were for BBO-MLP 0.842, 0.834, 0.809, 0.804, 0.801, 0.798, 0.792, 0.788, 0.773, and 0.743 and BSA-MLP 0.813, 0.805, 0.804, 0.803, 0.799, 0.799, 0.794, 0.795, 0.789, and 0.789, respectively. A 150-person swarm size characterizes the best-fit hybrid model for predicting landslide susceptibility mapping, and it belongs to the BBO-MLP model. According to the findings, these algorithms functioned well to enhance the MLP's learning potential. Additionally, adding metaheuristic algorithms such as BBO may significantly enhance the performance of the ANN with a drop in prediction MSE of 1.230 to 0.551 percent. Three well-known accuracy criteria—MSE, RMSE, and AUROC—were employed to create a rating system that compared the practicality of the utilized model. The result of this part showed that effectively the BBO & BSA algorithms increase the MLP's capacity for learning. |
|
|||
|
8 |
The Scoring system needs to be explained in detail. |
Thank you for your valuable comments. Kindly be noted that below text are added to the section 4 in order to provide more illustrations for the used scoring system. This systems are used in many other literature review too but we also believed that is the best to be discussed here too.
“The best predictive network results from the model with the highest score (or least rank in Table 2). It is noteworthy that, the scores came directly from the model prediction result accuracies. For instance, the lowest RMSE obtained results in a higher score for the specified model. However, for the R2, the higher R2 will result in a higher score. Therefore, the next sections make use of the results of these networks. Figures 7 and 8 further show how the MSE changes when the amount of each neuron per hidden layer increases or decreases. Table 2: A sensitivity study of forecasting landslide susceptibility mapping's number on varying numbers of neurons. |
|

Round 2
Reviewer 1 Report
Comparing with the first version, this time the authors increased some martarial to the introduction and the conclusions. Anyway, there still questions list below.
1) line 72, landslide displacement forecasting models,......actually most models were used for landslide susceptibility analysis, little of them can predict the slope displacement.
2) As line 115 say, the efficiency of the algorithm is the focus, the conclusions just tell the reader that the BBO-MLP takes less time to train. this is too short.
3) The final susceptibility map maybe not very good, the north part occur some landslide but its susceptibility level is low to very low. The northwest part and southeast part have some isolated circles, it must be some kind of problems. The authors should check the process or give some explain.
4) the revised version still not explain the physical meaning of population size or swarm size. Although the deep learning is difficult to explain, the author should try to make it much acceptable for more people.
Author Response
Title: Multilayer perceptron and their comparison with two nature-inspired
hybrid techniques of biogeography-based optimization (BBO) and backtracking
search algorithm (BSA) for assessment of Landslide susceptibility”
Journal: Land, Innovations – Data and Machine Learning
Manuscript ID: land-2112773
Many thanks for the valuable and constructive comments of the reviewers. An attempt was made to accurately apply all comments to the text of the article. Revisions made to the text of the article include:
1) Line 72, landslide displacement forecasting models ...actually most models were used for landslide susceptibility analysis, little of them can predict the slope displacement.
Thanks for the opinion of the respected judge. You are absolutely right, we forgot to correct this point in the previous version, and this keyword was corrected in the text of the article, and this point was very important, and it was corrected as follows:
With the ongoing updating of AI systems, certain nonlinear landslide susceptibility analysis models have been built. (Please see Lines 44-72)
2) As line 115 say, the efficiency of the algorithm is the focus, the conclusions just tell the reader that the BBO-MLP takes less time to train. This is too short.
Thanks to the valuable comments of the reviewer, the result was corrected and revised in the text of the article were made as follows:
The BBO algorithm enhances the accuracy of the MLP. Moreover, referring to the calculated total ranking scores of 10, 7, and 5, it becomes apparent that the BBO performs more efficiently than BSA in optimizing the MLP. However, the highest prediction accuracy is found in structures with less error and in this research, each of the two algorithms achieved a lower error (RMSEBBO-MLP & BSA-MLP = 0.551 and 0.557). Also, despite the superiority of the BBO-MLP in learning landslide patterns, both ensembles presented a close prediction accuracy (AUCBBO-MLP = 0.842 and AUCBSA-MLP = 0.771). In general, this BBO-MLP is more effective in improving neural network performance in this paper. (Please see Lines 613-621)
3) The final susceptibility map maybe not very good, the north part occur some landslide but its susceptibility level is low to very low. The northwest part and southeast part have some isolated circles, it must be some kind of problems. The authors should check the process or give some explain.
Thanks to the respected referee, your comments were very helpful. In order to clear the ambiguity, we have shown the landslide points in the current state on the maps. Also, we edited the maps and also outlined the necessary explanations in the text of the paper as follows:
The northern part of the study area in western Iran has fewer landslides and its sensitivity to landslides is low. But the northwest and southeast part of the study area in Kurdistan province is by mountainous areas and has a high potential for the risks of landslides, especially in the rainy season. As can be seen in the map, circular points show the landslide points in the status quo, which adapt to high-risk and high-risk classes indicating the high accuracy of the two algorithms used for optimization. Other areas that are classified in the northern part of the moderate risk indicate that according to the model forecasts these areas also have the potential for landslides and should be taken into account in crisis management planning. (Please see Lines 560-568 and Figure 13).
4) The revised version still not explain the physical meaning of population size or swarm size. Although the deep learning is difficult to explain, the author should try to make it much acceptable for more people.
Answer: Thank you very much for your comments. One of the best and recently published sources that can help readers to find a deeper understanding on the abovementioned subjects is https://link.springer.com/book/10.1007/978-3-031-09835-2. However, the below texts is the simplest and most helpful text that we could provide for the terms used in this study.
Due to model high complexity, researchers tent to provide an optimized solutions. Optimization is finding the optimum values of a problem's variables to minimize or maximize an objective function. In other word, optimization is finding the best solution to a trial by adjusting the importance of the variables that impact the result [12]. Optimization aims to minimize or maximize an objective function, a mathematical expression that reflects the item being improved. For example, suppose a company wants to make the most money possible. In that case, the objective function could be a mathematical model of the company's profit based on many factors, such as production levels, prices, and advertising costs. By adjusting these variables and finding the values that maximize the objective function, the company may find the most feasible solution to the problem of maximizing profits.
Due to the improvement of numerous solutions during optimization, multi-solution-based algorithms have a more significant local optimum avoidance by nature. In this case, more solutions may allow a solution trapped in a local optimum to escape from it. Multiple-solution-based algorithms examine a more significant section of the search region than single-solution-based algorithms; hence, the likelihood of getting the global optimum is greater [12, 13]. In addition, information about the search space may be shared across several solutions, which expedites progress toward the ideal. Despite their many benefits, multi-solution-based algorithms need additional function assessments. The most popular single-solution-based algorithms are hill climbing and simulated annealing. Both algorithms are based on a similar premise, but stochastic cooling allows SA to avoid local optimums more effectively [12].
Iterated Local Search (ILS) [14] and Tabu Search (TS) [15] are two modern algorithms based on a single solution. Popular multi-solutions-based algorithms [15-17] include genetic algorithms (GA), particle swarm optimization (PSO), ant colony optimization (ACO), and differential evolution (DE). Darwin's evolution of natural selection impacted the design of the GA algorithm. This algorithm sees solutions as solutions, and their parameters reflect their DNA. This algorithm is primarily motivated by natural selection, with the best individuals preferring to contribute more to improving mediocre solutions.
The GA algorithm represents solutions to a problem as "individuals" in a “population”, with the parameters of each individual (its "genes") reflecting different components of the solution. The GA algorithm employs concepts that imitate the process of natural evolution, such as selection, crossover (recombination), and mutation, to a population of individuals. A genetic algorithm seeks to find the ideal solution to a problem by "evolving" a population of solutions over time. The concept of "survival of the fittest" is implemented in the GA algorithm via the selection process, in which the best individuals have a greater chance of being picked to participate in the evolution of the population. This guarantees that the population improves over time since the best solutions will likely be passed on to future generations. The PSO algorithm resembles the feeding behavior of flocks of birds and schools of fish. This algorithm improves solutions relative to the best solutions previously reached by each particle and the best solution improved by the “swarm”. The ACO algorithm imitates the collective behavior of ants in finding the shortest route from the nest to the food source. DE employs simple formulae that incorporate the parameters of previous solutions to expand the candidate pool for a specific problem.
Two features distinguish the two types of nature-inspired algorithms [18]: enhancing solutions until they fulfill end criteria and splitting the optimization process into two parts, exploration and exploitation. Exploration is an algorithm's propensity to display highly unexpected behavior, resulting in significantly different solutions. Significant differences in the solutions motivate a deeper exploration of the search space and, therefore, the identification of its most promising sections. As an algorithm leans toward exploitation, solutions often encounter changes on a smaller scale and prefer to explore them locally. A combination of exploration and exploitation may lead to discovering the optimal global solution for a particular optimization problem.
(Please see section introduction in lines 74-137)

Reviewer 2 Report
This article can be accepted.
Author Response
thanks very much.